# Finding Second-Order Stationary Points Efficiently in Smooth Nonconvex Linearly Constrained Optimization Problems

**Songtao Lu**
IBM Thomas J. Watson Research Center
Yorktown Heights, NY 10598, USA
songtao@ibm.com

**Meisam Razaviyayn**
University of Southern California
Los Angeles, CA 90089, USA
razaviya@usc.edu

**Bo Yang** *
Amazon Alexa
Cambridge, MA 02142, USA
amzbyang@amazon.com

**Kejun Huang**
University of Florida
Gainesville, FL 32611, USA
kejun.huang@ufl.edu

**Mingyi Hong**
University of Minnesota
Minneapolis, MN 55455, USA
mhong@umn.edu

## Abstract

This paper proposes two efficient algorithms for computing approximate second-order stationary points (SOSPs) of problems with *generic* smooth non-convex objective functions and *generic* linear constraints. While finding (approximate) SOSPs for the class of smooth non-convex linearly constrained problems is computationally intractable, we show that *generic problem instances* in this class can be solved efficiently. Specifically, for a *generic* problem instance, we show that certain *strict complementarity* (SC) condition holds for all Karush-Kuhn-Tucker (KKT) solutions. Based on this condition, we design an algorithm named $\underline{S}$uccessive $\underline{N}$egative-curvature gr$\underline{A}$dient $\underline{P}$rojection (SNAP), which performs either conventional gradient projection or some negative curvature based projection steps to find SOSPs. SNAP is a second-order algorithm that requires $\widetilde{\mathcal{O}}(\max\{1/\epsilon_G^2, 1/\epsilon_H^3\})$ iterations to compute an $(\epsilon_G, \epsilon_H)$-SOSP, where $\widetilde{\mathcal{O}}$ hides the iteration complexity for eigenvalue-decomposition. Building on SNAP, we propose a first-order algorithm, named SNAP$^+$, that requires $\mathcal{O}(1/\epsilon^{2.5})$ iterations to compute $(\epsilon, \sqrt{\epsilon})$-SOSP. The per-iteration computational complexities of our algorithms are *polynomial* in the number of constraints and problem dimension. To the best of our knowledge, this is the first time that first-order algorithms with polynomial per-iteration complexity and global sublinear rate are designed to find SOSPs of the important class of non-convex problems with linear constraints (almost surely).

## 1   Introduction

We consider the following class of non-convex linearly constrained optimization problems

$$\underset{\mathbf{x}}{\text{minimize}} \quad f(\mathbf{x}), \quad \text{subject to } \mathbf{x} \in \mathcal{X} \triangleq \{\mathbf{x} \mid \mathbf{A}\mathbf{x} \leq \mathbf{b}\} \tag{1}$$

where $f : \mathbb{R}^d \to \mathbb{R}$ is twice differentiable (possibly non-convex); $\mathbf{A} \in \mathbb{R}^{m \times d}$ and $\mathbf{b} \in \mathbb{R}^m$ are some given matrix and vector. Such a class of problems finds many applications in machine learning and data science. For example, in the nonnegative matrix factorization (NMF) problem, a given data

matrix $\mathbf{M} \in \mathbb{R}^{n \times p}$ is to be factorized into two nonnegative matrices $\mathbf{W} \in \mathbb{R}^{n \times k}$ and $\mathbf{H} \in \mathbb{R}^{p \times k}$ such that $\|\mathbf{W}\mathbf{H}^T - \mathbf{M}\|_F^2$ is minimized [1]. Further, for non-convex problems with $\ell_1$ regularizer (such as sparse PCA [2]),we need to solve $\min \ g(\mathbf{x}) + \|\mathbf{x}\|_1$, which can be equivalently written as

$$\min \ g(\mathbf{x}) + \mathbf{1}^T \mathbf{y}, \quad \text{subject to } -\mathbf{y} \leq \mathbf{x} \leq \mathbf{y}. \tag{2}$$

Applications having these linear inequality constraints include neural networks training with the nonnegative constraint [3], nonnegative tensor factorization [4], statistical learning with simplex constraints [5], and portfolio optimization under budget constraints [6], to name just a few.

Recently, algorithms that can escape strict saddle points (stationary points at which there exist directions of negative curvature) for unconstrained non-convex problems have attracted considerable research attention, and they find applications in tensor decomposition [7], phase retrieval [8], low-rank matrix factorization [9], etc. However, it is not straightforward to extend these "saddle-point escaping" algorithms to problems with simple constraints or non-smooth regularizers. As will be seen shortly, even *checking* that a solution is a second-order stationary point (SOSP) for linearly constrained problems can be daunting. The main task of this paper is to identify situations in which finding a SOSP for problem (1) is easy and to design efficient algorithms for this task.

**Related work**. For unconstrained smooth problems, there has been a line of work that develops algorithms by using both the gradient direction and negative curvature so that SOSPs can be attained within a certain number of iterations [10, 11, 12]. For example, based on the fast top eigenvector computation, an accelerated gradient method [11] is guaranteed to converge to SOSPs with a quantifiable rate. By exploiting the eigendecomposition, the convergence rate of some variants of the Newton method to SOSPs has been shown in [13, 14, 15, 16], where the algorithm is assumed to be able to access the (inexact) Hessian matrix around strict saddle points. Another way of finding negative curvature is to occasionally add noise to the iterates. Algorithms in this class include a perturbed version of (accelerated) gradient descent (GD) [17, 18] and NEgative-curvature-Originated-from-Noise (NEON) [19], and their extensions [19, 20, 21, 22]. In practice, there may be some constraints, such as equality and inequality constraints. For the equality constrained case (e.g., linear/manifolds constraints), negative curvature methods, including both first- and second-order ones, have been proposed for which SOSPs can be obtained either asymptotically [23, 24] or with provable convergence rates [25, 26] under some mild conditions.

Despite these recent exciting developments, the aforementioned methods are unable to incorporate even the simplest linear *inequality* constraints. Existing methods either directly rely on *second-order* descent directions of the objective function [27, 10], or use this information together with other methods such as the trust region method [28], cubic regularized Newton's method [29], primal-dual algorithm [30], etc. These algorithms are generally undesirable for large-scale problems due to the scalability issues when computing the second-order information. However, to the best of our knowledge, there has been no first-order method that can provably compute SOSPs for linearly constrained problem (1).

An even more challenging issue is that finding SOSP for general linearly constrained non-convex problems (1) is NP-hard. Although some existing algorithms are shown with being able to find SOSPs for a certain class of constrained nonconvex problems [31, 32], their iteration complexities are still unknown. Indeed, it has been shown in [33, 34] that even obtaining the approximate SOSPs is hard in terms of both the total number of constraints and the inverse of the desired second-order optimality error. So existing methods for finding SOSPs with global convergence rate all require exponential complexity (exponential in the total number of constraints). To be more specific, while the algorithm in [35] is presented for solving a general type of constrained problems to SOSPs, the authors noted (by utilizing results from [36]) that the subroutine in their algorithm is NP-hard for (1). Therefore, subsequently the authors only focused on solving some special quadratically constrained problems. The work of [33] also indicated that [35] cannot be applied to linearly constrained problems. Similarly, the algorithms in [33, 34] are only applicable in the presence of a small number of linear constraints and do not scale polynomially as the number of constraints increases. Therefore, even for the relatively simple NMF problem, solving subproblems in [33, 34, 35] requires exponential complexities in the number of constraints.

Different from [33, 34, 35], instead of focusing on developing algorithms that work for all problem instances, in this work we design algorithms for *almost all* linearly constrained problems. Our results show that under strict complementarity condition (which we show it holds with probability one in many cases) escaping saddle points efficiently is in fact possible. More precisely, the key difference between our work and the existing ones is fundamental and relies upon distinct definitions/views.

Our approach is built upon a relatively easy-to-find SOSP notion, then we show that this notion is *almost surely equivalent* to the ones used in [33, 34, 35] when the problem data is random. Also, in contrast to these works [33, 34, 35], this is the only paper that provides multiple numerical experiments show the effectiveness of exploring the directions of negative curvature. This is partly due to the fact that the previously proposed algorithms are difficult to implement in many cases which involve solving NP-hard problems per-iteration.

**Contributions of this work**. We first introduce two notions of (approximate) SOSPs for problem (1), one is based on identifying the active constraints at a given solution (referred to as SOSP1), and the other is based on the feasible directions orthogonal to the gradient (referred to as SOSP2) [33, 35, 37]. In particular, we show that, these two conditions become *equivalent* when a certain (provably mild) strict complementarity (SC) condition is satisfied. Such equivalence condition enables us to design an algorithm, named Successive Negative-curvature grAdient Projection (SNAP), which exploits the active sets of the solution path, and is computationally much simpler compared with existing methods based on checking feasible directions orthogonal to gradient. Further, we extend SNAP by proposing a first-order algorithm (abbreviated as SNAP$^+$) which only utilizes gradient steps to extract the directions of negative curvature.

The main contributions of this work are as follows:
▶ We analyze the equivalence of two different notions of (approximate) SOSPs for (1), under the strict complementarity condition. The analysis provides new insights of solution structures, and has proven useful in the subsequent algorithm design.
▶ We propose the SNAP algorithm, which requires $\mathcal{O}(\max\{1/\epsilon_G^2, 1/\epsilon_H^3\})$ iterations to compute an $(\epsilon_G, \epsilon_H)$-SOSP1, and its per-iteration complexity is polynomial in $(d, m)$, as long as the projection, gradient, Hessian, and eigenvalue decomposition can be computed efficiently.
▶ We extend SNAP to SNAP$^+$, an algorithm that only uses the gradient of the objective function, while being able to compute $(\epsilon, \sqrt{\epsilon})$-SOSP1s for problem (1) within $\mathcal{O}(1/\epsilon^{2.5})$ iterations. Each iteration of the algorithm only involves vector operations and projections to the feasible set, which can be done with a polynomial complexity under reasonable oracles [38]. To the best of our knowledge, this is the first first-order method that is capable of achieving the above stated properties.

**Notation.** Bold lower case characters, e.g., $\mathbf{x}$ represents vectors and bold capital ones, e.g., $\mathbf{X}$ denote matrices; $\mathbf{x}_i$ or $(\mathbf{A}\mathbf{x})_i$ denotes the $i$th entry of vector $\mathbf{x}$ or $\mathbf{A}\mathbf{x}$. $\mathbf{A}^\dagger$ denotes the pseudo-inverse of matrix $\mathbf{A}$, and $\|\mathbf{A}\|$ denotes the spectral norm of $\mathbf{A}$.

## 2 Preliminaries

We make the following blanket assumption for problem (1).

**Assumption 1.** $f(\mathbf{x})$ *is* $L_1$-*gradient Lipschitz and* $L_2$-*Hessian Lipschitz, i.e.,* $\|\nabla f(\mathbf{x}) - \nabla f(\mathbf{y})\|_2 \leq L_1\|\mathbf{x} - \mathbf{y}\|$, $\|\nabla^2 f(\mathbf{x}) - \nabla^2 f(\mathbf{y})\|_2 \leq L_2\|\mathbf{x} - \mathbf{y}\|$, $\forall \mathbf{x}, \mathbf{y} \in \mathcal{X}$.

The following definitions will be useful. Let $\mathcal{A}(\mathbf{x}) = \{j \mid \mathbf{A}_j\mathbf{x} = \mathbf{b}_j, \forall j \in [m]\}$ denote the active set at $\mathbf{x} \in \mathcal{X}$, where $\mathbf{A}_j$ denotes the $j$th row of matrix $\mathbf{A}$ and $\mathbf{b}_j$ denotes the $j$th entry of $\mathbf{b}$. Define $\overline{\mathcal{A}(\mathbf{x})}$ to be the complement of the set $\mathcal{A}(\mathbf{x})$, i.e.,

$$\overline{\mathcal{A}(\mathbf{x})} \cup \mathcal{A}(\mathbf{x}) = [m], \quad \overline{\mathcal{A}(\mathbf{x})} \cap \mathcal{A}(\mathbf{x}) = \emptyset. \tag{3}$$

Stacking the *active* constraints at $\mathbf{x}$, we define

$$\mathbf{A}_{\mathcal{A}}(\mathbf{x}) \triangleq [ \ \ldots \ \ \mathbf{A}_j \ \ \ldots \ ]^T \in \mathbb{R}^{|\mathcal{A}(\mathbf{x})| \times d}, \quad j \in \mathcal{A}(\mathbf{x}). \tag{4}$$

Similarly, we can define $\mathbf{b}_{\mathcal{A}}(\mathbf{x}) \in \mathbb{R}^{|\mathcal{A}(\mathbf{x})| \times 1}$ by stacking the entries of $\mathbf{b}$ corresponding to the active set of constraints. At a given point $\mathbf{x}$, we define the projection onto the space orthogonal to the space spanned by the gradients of the active constraints as

$$\pi_{\mathcal{A}^\perp}(\mathbf{x}) \triangleq \mathbf{P}(\mathbf{x})\mathbf{x}, \quad \text{where } \mathbf{P}(\mathbf{x}) \triangleq \left(\mathbf{I} - \mathbf{A}_{\mathcal{A}}^T(\mathbf{x})\left(\mathbf{A}_{\mathcal{A}}(\mathbf{x})\mathbf{A}_{\mathcal{A}}^T(\mathbf{x})\right)^\dagger \mathbf{A}_{\mathcal{A}}(\mathbf{x})\right). \tag{5}$$

Here, $\mathbf{P}(\mathbf{x})$ represents the projection matrix to the null space of $\mathbf{A}_{\mathcal{A}}(\mathbf{x})$. Define $\mathbf{P}_\perp(\mathbf{x})$ as the projector to the column space of $\mathbf{A}_{\mathcal{A}}^T(\mathbf{x})$. Similarly, let us define

$$q_\pi(\mathbf{x}) \triangleq \pi_{\mathcal{A}^\perp}(\nabla f(\mathbf{x})) = \mathbf{P}(\mathbf{x})\nabla f(\mathbf{x}). \tag{6}$$

To define the first-order optimality gap, let us introduce the *proximal gradient* as: $g_\pi(\mathbf{x}) \triangleq (1/\alpha)(\pi_{\mathcal{X}}(\mathbf{x} - \alpha\nabla f(\mathbf{x})) - \mathbf{x})$, with $\pi_{\mathcal{X}}(\mathbf{v}) \triangleq \arg\min_{\mathbf{w} \in \mathcal{X}} \|\mathbf{w} - \mathbf{v}\|^2$, where $\alpha > 0$ is a

given constant and $\pi_{\mathcal{X}}$ denotes the projection operator onto the feasible set. Then $\|g_\pi(\mathbf{x})\|$ can be used to define the *first-order optimality gap* for any $\mathbf{x} \in \mathcal{X}$.

First, by the popular definition of second-order necessary conditions for local minimum solutions [37, Proposition 3.3.1], we can define the *exact* SOSP as below.

**Definition 1.** *The point $\mathbf{x}^* \in \mathcal{X}$ is a SOSP of* (1) *if*

$$\|g_\pi(\mathbf{x}^*)\| = 0, \qquad\qquad \text{(first-order condition)} \qquad (7a)$$

$$\mathbf{y}^T \nabla^2 f(\mathbf{x}^*)\mathbf{y} \geq 0, \quad \forall\, \mathbf{y} \text{ satisfying } \mathbf{A}_{\mathcal{A}}(\mathbf{x}^*)\mathbf{y} = 0, \qquad \text{(second-order condition)} \qquad (7b)$$

*where $\mathbf{A}_{\mathcal{A}}(\mathbf{x}^*)$, defined in* (4)*, is a matrix that collects the active constraints.*

Accordingly, we can define the *approximate* SOSPs and first-order stationary points (FOSPs), respectively.

**Definition 2.** *A point $\mathbf{x}^* \in \mathcal{X}$ is an $(\epsilon_G, \epsilon_H)$-SOSP1 of* (1) *if the following conditions hold*

$$\|g_\pi(\mathbf{x}^*)\| \leq \epsilon_G, \qquad\qquad \text{(approx. 1st-order condition)} \qquad (8a)$$

$$\mathbf{y}^T \nabla^2 f(\mathbf{x}^*)\mathbf{y} \geq -\epsilon_H, \forall \mathbf{y} \text{ s.t. } \mathbf{A}_{\mathcal{A}}(\mathbf{x}^*)\mathbf{y} = 0, \|\mathbf{y}\| = 1. \qquad \text{(approx. 2nd-order condition)} \qquad (8b)$$

**Definition 3.** *A point $\mathbf{x}^* \in \mathcal{X}$ is an $\epsilon_G$-first-order stationary point ($\epsilon_G$-FOSP1) of the first kind (FOSP1), if it satisfies $\|g_\pi(\mathbf{x}^*)\| \leq \epsilon_G$.*

It is clear that by using the definition (5), we can rewrite condition (8b) as

$$\lambda_{\min}(\mathbf{H}_{\mathbf{P}}(\mathbf{x}^*)) \geq -\epsilon_H, \quad \text{with} \quad \mathbf{H}_{\mathbf{P}}(\mathbf{x}^*) := \mathbf{P}(\mathbf{x}^*)\nabla^2 f(\mathbf{x}^*)\mathbf{P}(\mathbf{x}^*), \qquad (9)$$

where $\lambda_{\min}(\cdot)$ is the operator that returns the smallest eigenvalue of a matrix.

*Remark 1.* For unconstrained problems, condition (8b) reduces to $\lambda_{\min}(\nabla^2 f(\mathbf{x}^*)) \geq -\epsilon_H$, since in the case $\mathbf{P}(\mathbf{x}) \equiv \mathbf{I}$ and $\mathcal{A}(\mathbf{x}) \equiv \emptyset$.

Note that the conditions in (7) and (8) are necessary for $\mathbf{x}^*$ to be a local minimum solutions. There are, of course, many other necessary conditions of this kind. Therefore, to distinguish from various solution concepts, we will refer to the solutions satisfying the above conditions as SOSP *of the first kind* and $(\epsilon_G, \epsilon_H)$-SOSP *of the first kind*, abbreviated as SOSP1 and $(\epsilon_G, \epsilon_H)$-SOSP1, respectively. Below we present another popular second-order solution concept; see [35, 33], and [37, Chapter 2].

**Definition 4.** *A point $\mathbf{x}^* \in \mathcal{X}$ is an $(\epsilon_G, \epsilon_H)$-second-order stationary point of the second kind ($(\epsilon_G, \epsilon_H)$-SOSP2), if the following conditions are satisfied:*

$$\nabla f(\mathbf{x}^*)^T(\mathbf{x} - \mathbf{x}^*) \geq -\epsilon_G, \quad \forall \mathbf{x} \in \mathcal{X}, \quad \text{with} \quad \|\mathbf{x} - \mathbf{x}^*\| \leq 1 \qquad (10a)$$

$$(\mathbf{x} - \mathbf{x}^*)^T \nabla^2 f(\mathbf{x}^*)(\mathbf{x} - \mathbf{x}^*) \geq -\epsilon_H, \quad \forall \mathbf{x} \in \mathcal{X} \quad \text{with} \quad \nabla f(\mathbf{x}^*)^T(\mathbf{x} - \mathbf{x}^*) = 0. \qquad (10b)$$

**Definition 5** ($\epsilon_G$-FOSP2)**.** *If a solution $\mathbf{x}^* \in \mathcal{X}$ only satisfies* (10a)*, we call it an $\epsilon_G$-first-order stationary point of the second kind (FOSP2).*

The classical result [36] shows that checking $(\epsilon_G, \epsilon_H)$-SOSP2 for (1) is NP-hard in the problem dimension and in $\log(1/\epsilon_H)$ even for the class of quadratic functions. This hardness result has recently been strengthened by showing NP-hardness in terms of problem dimension and in $1/\epsilon_H$ [33]. On the other hand, checking $(\epsilon_G, \epsilon_H)$-SOSP1 condition only requires projection onto linear subspaces and finding minimum eigenvalues. A natural question then arises: How do these different kinds of *approximate* and *exact* SOSP concepts relate to each other? In what follows, we provide a concrete answer to this question. This answer relies on a critical concept called *strict complementarity*, which is introduced below.

**Definition 6.** *A given primal-dual pair $(\mathbf{x}^*, \boldsymbol{\mu}^*)$ for problem* (1) *satisfies the Karush-Kuhn-Tucker (KKT) condition with strict complementarity (SC) if*

$$\nabla f(\mathbf{x}^*) + \sum_{j=1}^{m} \boldsymbol{\mu}_j^* \mathbf{A}_j^T = 0, \qquad (11a)$$

*for each $j \in [m]$, either $\boldsymbol{\mu}_j^* > 0$, $\mathbf{A}_j \mathbf{x}^* = \mathbf{b}_j$ or $\boldsymbol{\mu}_j^* = 0$, $\mathbf{A}_j \mathbf{x}^* < \mathbf{b}_j$.* $\qquad (11b)$

Note that the SC condition has been assumed and used in convergence analysis of many algorithms, e.g., trust region algorithms for non-convex optimization with bound constraints in [39, 40, 41]. The results below extend a recent result in [42, Proposition 2.3], which shows that SC is satisfied for box constrained non-convex problems.

**Proposition 1.** *Suppose $f(\mathbf{x}) = g(\mathbf{x}) + \mathbf{q}^T\mathbf{x}$ in problem* (1) *where $g(\mathbf{x})$ is differentiable. Let $\mathbf{x}^*$ be a KKT point of problem* (1)*. If $\mathbf{q}$ is generated from a continuous measure, and if the vectors $\{\mathbf{A}_j \mid j \in \mathcal{A}(\mathbf{x}^*)\}$ are linearly independent, then the SC condition holds with probability one (w.p.1).*

The following result can be shown using similar techniques as Proposition 1. for details.

**Corollary 1.** *Suppose $f(\mathbf{x}) = g(\mathbf{x}) + \mathbf{q}^T\mathbf{x}$ and $g(\mathbf{x})$ is differentiable. If $(\mathbf{q}, \mathbf{b})$ is generated from a continuous measure, then the SC condition holds for* (1) *w.p.1.*

These results show that for a certain generic choice of objective functions, the SC condition is satisfied. As we will see in the next section, this SC condition leads to the equivalence of SOSP1 and SOSP2 conditions. Hence, instead of working with the computationally intractable SOSP2 condition, we can use the tractable SOSP1 condition for developing algorithms. In other words, by adding a small random linear perturbation to the objective function, which does not practically change the landscape of the optimization problem, we can avoid the computational intractability of SOSP2.

## 3 Almost Sure Equivalence of SOSP1 and SOSP2

To understand the relation between SOSP1 and SOSP2, let us consider the following example.

*Example 1.* Consider the following box constrained quadratic problem:

$$\underset{x_1,x_2}{\text{minimize}} \quad -x_1^2 - x_2^2, \quad \text{subject to} \quad 0 \le x_1 \le 1, \quad 0 \le x_2 \le 1. \tag{12}$$

Clearly the point $\mathbf{x}^* = (0,0)$ is an SOSP1. However, the point $\mathbf{x}^* = (0,0)$ is *not* an SOSP2 according to the definition in (10). This is because the condition $\nabla f(\mathbf{x}^*)^T(\mathbf{x} - \mathbf{x}^*) = 0$ is true for all feasible $\mathbf{x}$, but $(\mathbf{x} - \mathbf{x}^*)^T \nabla^2 f(\mathbf{x}^*)(\mathbf{x} - \mathbf{x}^*) = -2$ for $\mathbf{x} = (1,1)$. ∎

The above example suggests that condition (10) is *stronger* than (8), even when $\epsilon_H = 0, \epsilon_G = 0$. Below, we show that these conditions become *equivalent* when the SC condition (11) holds true. The equivalence between the first-order optimality conditions (8a) and (10a) is obvious.

**Proposition 2.** *Suppose that every KKT point $(\mathbf{x}^*, \boldsymbol{\mu}^*)$ of problem* (1) *satisfies* (11)*. Then the conditions* (8) *and* (10) *are equivalent for the case of $\epsilon_H = \epsilon_G = 0$. In other words, for any $\mathbf{x}^* \in \mathcal{X}$, if it satisfies* (8) *for $\epsilon_H = \epsilon_G = 0$, then it also satisfies* (10) *for $\epsilon_H = \epsilon_G = 0$, and vice versa.*

Next we establish the relations between *approximate* second-order conditions.

**Corollary 2.** *The second-order conditions* (8b) *and* (10b) *are equivalent in the following sense: suppose the SC condition* (11) *holds, then any $(0, \epsilon_H)$-SOSP2 must satisfy $(0, \epsilon_H)$-SOSP1, and vice versa.*

Although at this point we have not shown the equivalence of $(\epsilon_G, \epsilon_H)$-SOSP1 and $(\epsilon_G, \epsilon_H)$-SOSP2 (a result that remains very challenging), we provide the following alternative result showing that $(\epsilon_G, \epsilon_H)$-SOSP1 is a valid approximation of $(0, 0)$-SOSP1, which in turn is equivalent to $(0, 0)$-SOSP2 by Proposition 2.

**Proposition 3.** *Let $\{\mathbf{x}^{(r)}\}_{r=1}^{\infty}$ be a sequence generated by an algorithm. Assume for each $r$, the point $\mathbf{x}^{(r)}$ be an $(\epsilon_G^{(r)}, \epsilon_H^{(r)})$-SOSP1. Assume further that $\{(\epsilon_G^{(r)}, \epsilon_H^{(r)})\}$ converges to $(0,0)$. Then any limit point of $\{\mathbf{x}^{(r)}\}_{r=1}^{\infty}$ is an exact SOSP1.*

We conclude this section by a few remarks comparing the two solution concepts:
**1)** For a given $\mathbf{x}$, checking whether SOSP1 holds is computationally tractable (since it only requires finding the active constraints, computing its null space, and performing an eigenvalue decomposition), while checking SOSP2 is in general NP-hard [33].
**2)** It is relatively easy to design an algorithm based on active constraints: When a sequence of iterates approaches an FOSP, the corresponding active set remains the same (see [43, Proposition 1.37], [44, Proposition 3]). Therefore locally the inequality constrained problem is reduced to an equality constrained problem, whose second-order conditions are much easier to satisfy; see [45][46].
**3)** Since the SC condition is satisfied with probability one for problems with random data, finding SOSP1 is already good enough for these problems.
**4)** The SOSP1 represents an interesting tradeoff between the quality of the solutions and computational complexity of the resulting algorithms. In what follows, we will design efficient algorithms that can compute SOSPs based on this solution concept.

# 4 The Proposed SNAP Algorithm for Computing SOSP1

**Algorithm Description.** Our proposed algorithm successively performs two main steps: a conventional projected gradient descent (PGD) step and a negative curvature descent (NCD) step. Assuming that the feasible set $\mathcal{X}$ is easy to project on (e.g., the non-negativity constraints for the NMF problem), the PGD finds an approximate first-order point efficiently, while the negative curvature descent step explores curvature around a first-order stationary point to move the iterates forward.

To provide a detailed description of the algorithm, we will first introduce the notion of *feasible directions* using the directions $\mathbf{y}$ in (7b) and (8b). In particular, for a given point $\mathbf{x} \in \mathcal{X}$, we define the feasible direction subspace as $\mathcal{F}(\mathbf{x}) = \mathsf{Null}(\mathbf{A}_\mathcal{A}(\mathbf{x}))$, where $\mathsf{Null}(\mathbf{A}_\mathcal{A}(\mathbf{x}))$ denotes the null space of matrix $\mathbf{A}_\mathcal{A}(\mathbf{x})$. Such directions are useful for extracting negative curvature directions. We will refer to the subspace $\mathcal{F}(\mathbf{x})$ as the *free space* and we refer to its orthogonal complement as the *active space*. The inputs in the algorithm are some constants related to the problem data, the initial point $\mathbf{x}^{(1)}$, and parameters $\epsilon_G, \epsilon_H$. Further $0 < \alpha_\pi < 1/L_1$ is the step-size, and $\delta > 0$ is the accuracy of the curvature finding algorithm, which will be determined later.

---

**Algorithm 1** Successive Negative-curvature grAdient Projection algorithm (SNAP)

---

1: **Input:** $\mathbf{x}^{(1)}, \epsilon_G, \epsilon_H, L_1, L_2, \alpha_\pi, \delta, \mathbf{A}, \mathbf{b}, r_{\mathsf{th}}, \mathsf{flag} = \Diamond, \mathsf{flag}_\alpha = \Diamond, r_{\mathsf{last}} = 0$
2: **for** $r = 1, \ldots$ **do**
3:     **if** $\|g_\pi(\mathbf{x}^{(r)})\| \leq \epsilon_G$ and ($\mathsf{flag}_\alpha = \Diamond$ or ($\mathsf{flag}_\alpha = \emptyset$ and $r - r_{\mathsf{last}} \geq r_{\mathsf{th}}$)) **then**
4:         $[\mathsf{flag}, \mathbf{v}(\mathbf{x}^{(r)}), -\epsilon'_H(\delta)] = $ *Negative-Eigen-Pair*$(\mathbf{x}^{(r)}, f, \delta, \mathcal{F}(\mathbf{x}^{(r)}))$
5:         **if** $\mathsf{flag} = $ *exists negative curvature* **then**
6:             Compute $q_\pi(\mathbf{x}^{(r)})$ by (6)
7:             Choose $\mathbf{v}(\mathbf{x}^{(r)})$ such that $q_\pi(\mathbf{x}^{(r)})^T\mathbf{v}(\mathbf{x}^{(r)}) \leq 0$
8:             **if** $\frac{3L_1\epsilon'_H(\delta)}{L_2}q_\pi(\mathbf{x}^{(r)})^T\mathbf{v}(\mathbf{x}^{(r)}) - \frac{135L_1\epsilon'^3_H(\delta)}{128L_2^2} \geq -\|q_\pi(\mathbf{x}^{(r)})\|^2$ **then**
9:                 $\mathbf{d}^{(r)} = -q_\pi(\mathbf{x}^{(r)})$                        ▷ Choose gradient direction
10:             **else**     $\mathbf{d}^{(r)} = \mathbf{v}(\mathbf{x}^{(r)})$             ▷ Choose negative curvature direction
11:             **end if**
12:             Update $(\mathbf{x}^{(r+1)}, \mathsf{flag}_\alpha)$ by Algorithm 2             ▷ Perform line search
13:             **if** $\mathsf{flag}_\alpha = \emptyset$ **then**
14:                 $r_{\mathsf{last}} \leftarrow r$
15:             **end if**
16:         **else**     Output $\mathbf{x}^{(r)}$
17:         **end if**
18:     **else**     Update $\mathbf{x}^{(r+1)}$ by (13)                          ▷ Perform PGD
19:     **end if**
20: **end for**

---

Notice that as long as the computation of gradient, Hessian, and projection can be done in a polynomial number of floating point operations, the computational complexity of SNAP per iteration becomes polynomial. For most practical problems, it is reasonable to assume that gradient and Hessian can be computed efficiently. In addition, projection to linear inequality constraints is well-studied and can be done polynomially under reasonable oracles [38]. Two major steps of the algorithms are discussed below.

*The PGD step (line 18).* The conventional PGD, given below, with a constant step-size $\alpha_\pi > 0$:
$$\mathbf{x}^{(r+1)} = \pi_\mathcal{X}(\mathbf{x}^{(r)} - \alpha_\pi \nabla f(\mathbf{x}^{(r)})). \tag{13}$$
The PGD guarantees that the objective value decreases so that the algorithm can achieve some approximate FOSPs, i.e., $\|g_\pi(\mathbf{x}^{(r)})\| \leq \epsilon_G$.

*Negative curvature descent (NCD, line 4-17).* After the completion of PGD, $\|g_\pi(\mathbf{x}^{(r)})\|$ is already small. Suppose that the $(\epsilon_G, \epsilon_H)$-SOSP1 solution has not been found yet. Then our next step is to design an update direction to increase $\lambda_{\min}(\mathbf{H_P}(\mathbf{x}^{(r)}))$. Towards this end, a NCD step will be performed (line 4–17), which constructs update directions that can exploit curvature information about the Hessian matrix at $\mathbf{x}^{(r)}$, while ensuring that the iterates stay in the feasible region.

**The NCD Step.** The NCD further contains the following sub-procedures.

*Extracting negative curvature (line 4).* Assume that (9) does not hold. First, a procedure *Negative-Eigen-Pair* is called, which exploits some second-order information about the function at $\mathbf{x}^{(r)}$, and

returns an approximate eigen-pair $\{\mathbf{v}(\mathbf{x}^{(r)}), -\epsilon'_H(\delta)\}$ of the Hessian $\nabla^2 f(\mathbf{x}^{(r)})$. Such an approximate eigen-pair should satisfy the following requirements (with probability at least $1 - \delta$):

[1] $\mathbf{v}(\mathbf{x}^{(r)}) \in \mathcal{F}(\mathbf{x}^{(r)})$ and $\|\mathbf{v}(\mathbf{x}^{(r)})\| = 1$;

[2] $\mathbf{v}(\mathbf{x}^{(r)})^T \nabla^2 f(\mathbf{x}^{(r)}) \mathbf{v}(\mathbf{x}^{(r)}) \leq -\epsilon'_H(\delta)$ for some $\epsilon'_H(\delta)$ where $\exists \gamma > 0$ such that $\gamma \epsilon'_H(\delta) \geq \epsilon_H$.

If $\{\mathbf{v}(\mathbf{x}^{(r)}), -\epsilon'_H(\delta)\}$ satisfy the above conditions, *Negative-Eigen-Pair* returns $\diamond$ or "*exists negative curvature*", otherwise, it returns $\emptyset$. As long as (9) holds, many existing algorithms can achieve the two conditions stated above in a finite number of iterations (e.g., the Lanczos method [47]). Note that, these methods typically require to evaluate the Hessian matrix or Hessian-vector products.

*Backtracking line search (Algorithm 2)* (See Appendix A.) One important sub-procedure uses a line search to determine the step-size so that a feasible iterate $\mathbf{x}^{(r+1)}$ can be generated. The key in this step is to ensure that, either the new iterate achieves some kind of "sufficient" descent, or it touches the boundary of the feasible set. We use $\mathsf{flag}_\alpha$ to denote whether the updated iterate touches a new boundary or not. Note that in Algorithm 2 there are two directions that will be used to update iterate $\mathbf{x}^{(r)}$ and both of them stay in the *free space*, i.e., $q_\pi(\mathbf{x}^{(r)}), \mathbf{v}(\mathbf{x}^{(r)}) \in \mathcal{F}(\mathbf{x}^{(r)})$. Therefore, after performing the line search, the dimension of the feasible directions will be non-increasing, i.e., $\dim(\mathcal{F}(\mathbf{x}^{(r+1)})) \leq \dim(\mathcal{F}(\mathbf{x}^{(r)}))$ (see the details of the proof of Lemma 7).

*Selecting an update direction (line 8).* The actual update direction we use is chosen between the direction $\mathbf{v}(\mathbf{x}^{(r)}) \in \mathcal{F}(\mathbf{x}^{(r)})$ found in the previous step, and a direction $q_\pi(\mathbf{x}^{(r)}) \in \mathcal{F}(\mathbf{x}^{(r)})$ computed by directly projecting $\nabla f(\mathbf{x}^{(r)})$ to the subspace of feasible directions. Note that whichever directions we choose, we will have $\mathbf{d}^{(r)} \in \mathcal{F}(\mathbf{x}^{(r)})$. The selection criteria, given in line 8 of Algorithm 1, is motivated by the following descent property:

**Lemma 1.** *If $\mathbf{x}^{(r)}$ is updated by Algorithm 2 (the line search) and $\alpha_{\max}^{(r)}$ is not chosen, then the minimum decrease of the objective value by choosing $\mathbf{d}^{(r)} = -q_\pi(\mathbf{x}^{(r)})$ is no less than the one by selecting $\mathbf{d}^{(r)} = \mathbf{v}(\mathbf{x}^{(r)})$, and vice versa.*

*Remark 2.* (**"Flags" used in the algorithm**) After each NCD step, if $\mathsf{flag}_\alpha = \emptyset$ (i.e., some sufficient descent is achieved), we perform $r_{th}$ iterations of PGD between two successive eigenvalue decomposition (EVD)s. This design improves the practical efficiency by striking the balance between objective reduction and computational complexity. In practice $r_{th}$ can be chosen as any constant number. When $r_{th} = 0$, SNAP becomes simpler and has the same order of convergence rate as the case where $r_{th} > 0$.

**Theoretical Guarantees.** The convergence analysis of the proposed SNAP is presented as follows.

**Theorem 1.** *Suppose Assumption 1 is satisfied. There exists a sufficiently small $\delta'$ such that the sequence $\{\mathbf{x}^{(r)}\}$ generated by Algorithm 1 satisfies optimality condition (8) in the following number of iterations, with probability at least $1 - \delta'$:*

$$\widetilde{\mathcal{O}}\left(\max\left\{\frac{L_1 \min\{d, m\}}{\epsilon_G^2}, \frac{L_2^2 \max\{r_{th}, \min\{d, m\}\}}{\epsilon_H^3}\right\}(f(\mathbf{x}^{(1)}) - f^\star)\right) \quad (14)$$

*where the randomness comes from the oracle Negative-Eigen-Pair, $f^\star \triangleq \min_{\mathbf{x} \in \mathcal{X}} f(\mathbf{x})$, and $\widetilde{\mathcal{O}}$ hides the number of iterations run by an oracle Negative-Eigen-Pair.*

*Remark 3.* The convergence rate of SNAP has the same order in $\epsilon_G, \epsilon_H, L_1, L_2$, compared with those proposed in [35] and [33] (which compute $(\epsilon_G, \epsilon_H)$-SOSP2s). However, it is important to note that the per-iteration complexity of SNAP is polynomial in both problem dimension and in number of constraints, while algorithms proposed in [35] and [33] have exponential per-iteration complexity.

*Remark 4.* In particular, SNAP needs $\widetilde{\mathcal{O}}(\min\{d, m\}/\epsilon^2)$ number of iterations to achieve an $(\epsilon, \sqrt{\epsilon})$-SOSP1 by just substituting $\epsilon_G = \epsilon, \epsilon_H = \sqrt{L_2 \epsilon}$ and $r_{th} \sim \mathcal{O}(L_1/\sqrt{L_2 \epsilon})$.

## 5 First-order Successive Negative-curvature Gradient Projection (SNAP$^+$)

We propose a variant of SNAP, named SNAP$^+$, featuring a subspace gradient descent from noise perturbation (SGDN) procedure (as listed in Algorithm 4), which can extract the negative curvature in a subspace. Our work is motivated by a number of recent works, which show that occasionally adding random noise to the iterates of (unconstrained) GD can help escape from saddle points efficiently [17, 7]. The main advantage of the proposed SNAP$^+$ as compared to SNAP, is that its practical complexity can be improved significantly since the procedure of finding the negative eigen-pair is implemented by a first-order method.

Table 1: Convergence rates comparison of algorithms to SOSPs, where $\widetilde{\mathcal{O}}(\cdot)$ hides per-iteration complexity.

| Algorithm | Complexity P-I[1] | Iterations | $(\epsilon_G, \epsilon_H)$-SOSP | Oracle |
|---|---|---|---|---|
| ESP[2] [35] | $\mathcal{O}(\mathbf{exp}(m))$ | $\widetilde{\mathcal{O}}(\max\{\epsilon_G^{-2}, \epsilon_H^{-3}\})$ | $(\epsilon_G, \epsilon_H)$-SOSP2 | Hessian |
| SO-LC-TRACE[3] [34] | $\mathcal{O}(\mathbf{exp}(m))$ | $\widetilde{\mathcal{O}}(\max\{\epsilon_G^{-3/2}, \epsilon_H^{-3}\})$ | $(\epsilon_G, \epsilon_H)$-SOSP2 | Hessian-Vector product |
| **SNAP** (This work) | $\mathcal{O}(\text{poly}(d,m))$ | $\widetilde{\mathcal{O}}(\max\{\epsilon_G^{-2}, \epsilon_H^{-3}\})$ | $(\epsilon_G, \epsilon_H)$-SOSP1 | Hessian-Vector product |
| **SNAP$^+$** (This work) | $\mathcal{O}(\text{poly}(d,m))$ | $\widehat{\mathcal{O}}(\epsilon^{-5/2})$ | $(\epsilon, \epsilon^{\frac{1}{2}})$-SOSP1 | Gradient |

[1]P-I denotes per-iteration; [2] ESP denotes the escape saddle point algorithm [35]; [3] SO-LC-Trace denotes the second-order-linear constrained-TRACE algorithm [34].

**Negative Curvature Extraction by Subspace Gradient Descent from Noise (SGDN).** In particular, the key idea of these perturbation schemes is to use the difference of the gradient successively [19], given below, to approximate the Hessian-vector product

$$\mathbf{z}^{(\tau+1)} = \mathbf{z}^{(\tau)} - \beta(q_\pi(\mathbf{x}^{(r)} + \mathbf{z}^{(\tau)}) - q_\pi(\mathbf{x}^{(r)})), \quad \text{for } \tau = 1, \dots, T. \tag{15}$$

Here $T$ is some properly selected constant, $\beta \leq 1/L_1$ is the step-size and the algorithm is initialized from a random vector $\mathbf{z}^{(1)} \in \mathcal{F}(\mathbf{x}^{(r)})$ drawn from a uniform distribution in the interval $[0, \mathscr{R}]$, where $\mathscr{R}$ is some constant. This process can be viewed as performing power iteration around the strict saddle point. The details of the algorithm is presented in Algorithm 4 in Appendix D.1.

**Theoretical Guarantees of SNAP$^+$.** Essentially, when SGDN stops, it produces a direction $\mathbf{z}$ that satisfies the requirements of the outputs for the *Negative-Eigen-Pair* oracle in Algorithm 1. It follows that the rate claimed in Theorem 1 still holds for SNAP$^+$.

**Corollary 3.** *(Convergence rate of SNAP$^+$) Suppose Assumption 1 is satisfied and SGDN with step-size less than $1/L_1$ is used to find the negative eigen-pair. Then, there exists a sufficiently small $\delta'$ and $\sqrt{\epsilon} \leq L_1$ such that the sequence $\{\mathbf{x}^{(r)}\}$ generated by Algorithm 1 finds an $(\epsilon, \sqrt{\epsilon})$-SOSP1 in the following number of iterations with probability at least $1 - \delta'$ (where $\widehat{\mathcal{O}}$ hides the polynomial in terms of $d$ and $1/\epsilon$)*

$$\widehat{\mathcal{O}}\left(\max\left\{\frac{L_1^2}{L_2^{1/2}\epsilon^{5/2}}, \frac{L_1}{\epsilon^2}\right\}\min\{d,m\}(f(\mathbf{x}^{(1)}) - f^\star)\right). \tag{16}$$

*Remark 5.* SGDN in SNAP$^+$ is not needed for every step. Also, the accelerated version of SGDN and PGD (e.g., by incorporating Nesterov acceleration technique) can be used such that we can have a faster convergence rate of SNAP$^+$.

*Remark 6.* Note that the total rate of SNAP$^+$ includes the number of iterations required by SGDN, so it is $\widetilde{\mathcal{O}}(1/\epsilon^{\frac{1}{2}})$ slower than the rate of SNAP. To conclude this section, we provide in Table 1 some key features of a number of closely related existing algorithms for finding SOSPs for (1). To our best knowledge, SNAP$^+$ is the first first-order algorithm that has the provable convergence rate to SOSP1s, and the per-iteration complexity is polynomial in both $d$ and $m$.

## 6 Numerical Results

We showcase the numerical advantages of SNAP and SNAP$^+$, compared with PGD, PGD with a line search (PGD-LS) and gradient based alternating minimization (Alt-Min) for NMF problems. The starting point for all the algorithms is $\mathbf{X}^{(1)} = c\pi_{\mathcal{A}\perp}([\mathbf{W}^{(1)}; \mathbf{H}^{(1)}])$, where $\mathbf{W}^{(1)}$ and $\mathbf{H}^{(1)}$ are randomly generated. Constant $c$ controls the distance between the initialization point and the origin. The cases of $c = 1$ and $c = 10^{-10}$ correspond to large and small initialization, respectively. The rationale for considering a small initialization is that for NMF problems, it can be easily checked that $(\mathbf{W} = \mathbf{0}, \mathbf{H} = \mathbf{0})$ is a saddle point. By initializing around this point, we aim at examining whether indeed the proposed SNAP and SNAP$^+$ are able to escape from this region. Other details of the experimental setup are shown in Section E in the Appendix. The results of this experiment are presented in Figure 1. It can be observed that SNAP and SNAP$^+$ converge much faster by leveraging the negative curvature in Figure 1(a). In Figure 1(b), and obtain loss values that are many orders of magnitude smaller than those obtained by PGD and PGD-LS, confirming that the proposed methods are able to escape from saddle points, while PGD and PGD-LS get trapped. More comparison with

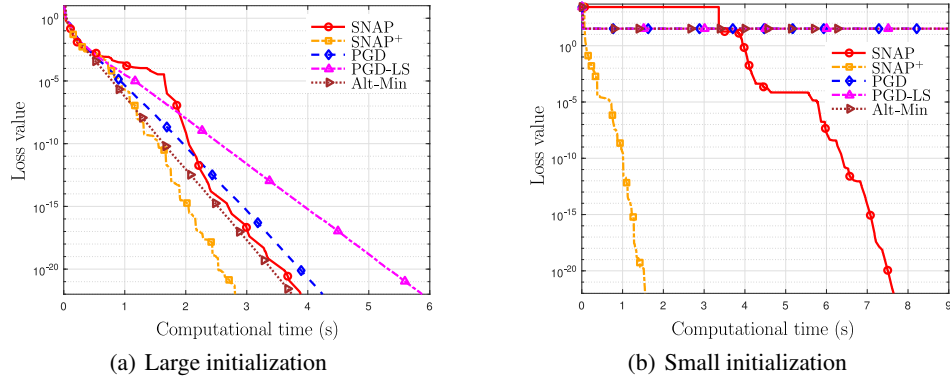

|                    |                    |
| :----------------: | :----------------: |
| (a) Large initialization | (b) Small initialization |

Figure 1: The computational time comparison of SNAP, SNAP$^+$, PGD, PGD-LS, and Alt-Min for NMF.

regard to loss values versus the number of iterations and computational time and other applications of algorithms (e.g., training nonnegative neural networks, penalized NMF, etc.) can be found in the Appendix E.

## 7 Broader Impact

Our main contributions in this work include both new theoretical and numerical results for solving nonconvex optimization problems under linear constraints. The theoretical part is regarding the new insight of a mathematical problem and the proposed algorithm is very general in the sense it can be applied not only to machine learning problems, but also to other general linear constrained problems in any other fields. Therefore, this works would be beneficial for both scientists/professors who are performing research in the area of machine learning and students who are studying operation research, engineering, data science, finance, etc. The theories and ideas in this work can potentially lead to significant improvements on the "off-the-shelf" optimization solvers and packages by equipping them with efficient modules for escaping saddle points in the presence of linear constraints. In addition to the methodological developments, the viewpoint of looking at the *generic* optimization problem instances could have potential broader impact on analyzing and resolving other issues in the continuous optimization field as well. While this work handles one specific hard task (i.e. finding SOSPs) by analyzing generic problem instances, this viewpoint could result in new tools and theories for dealing with other hard tasks for *generic* optimization instances.

We haven't found any negative impact of this work on both ethical aspects and future societal consequences.

## 8 Acknowledgment

Mingyi Hong is supported by NSF under Grant No. CMMI-1727757.

## Footnotes

*contributed to this work when he was working as a Ph.D. student at the University of Minnesota.

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
