[Supplementary Material]

# Appendices for "Finding Second-Order Stationary Points Efficiently in Smooth Nonconvex Linearly Constrained Optimization Problems"

## A  Details of Implementation of Algorithms

In this section, we will elaborate more about the ideas of designing SNAP.

First, we give the main motivation of selecting the update directions.

### A.1  Proof of Lemma 1

*Proof.* Suppose $-q_\pi(\mathbf{x}^{(r)})$ is chosen, and if $\alpha_{\max}^{(r)}$ in Algorithm is not chosen (i.e., if line 7 of Algorithm 2 does not hold true). Then, by the $L_1$-Lipschitz continuity, we have

$$f(\mathbf{x}^{(r)} + \alpha \mathbf{d}^{(r)}) \le f(\mathbf{x}^{(r)}) + \alpha q_\pi(\mathbf{x}^{(r)})^T \mathbf{d}^{(r)} + \frac{\alpha^2 L_1}{2} \|\mathbf{d}^{(r)}\|^2, \tag{17}$$

since $\nabla f(\mathbf{x}^{(r)}) = \mathbf{P}(\mathbf{x}^{(r)}) \nabla f(\mathbf{x}^{(r)}) + \mathbf{P}_\perp(\mathbf{x}^{(r)}) \nabla f(\mathbf{x}^{(r)})$ and $q_\pi(\mathbf{x}^{(r)}) = \mathbf{P}(\mathbf{x}^{(r)}) \nabla f(\mathbf{x}^{(r)})$. Therefore, according to (17) and (58) in the proof of Lemma 5, the minimum decrease of the objective is $d_q \triangleq -\frac{3}{8L_1} \|q_\pi(\mathbf{x}^{(r)})\|^2$. This is because the line search algorithm will terminate for some $\alpha \ge 1/(2L_1)$ if $-q_\pi(\mathbf{x}^{(r)})$ is chosen; see the proof of Lemma 5 for details.

Further, if $\mathbf{v}(\mathbf{x}^{(r)})$ is chosen and $\alpha_{\max}^{(r)}$ in Algorithm is not chosen, by $L_2$-Lipschitz continuity, we have

$$f(\mathbf{x}^{(r)} + \alpha \mathbf{d}^{(r)}) \le f(\mathbf{x}^{(r)}) + \alpha q_\pi(\mathbf{x}^{(r)})^T \mathbf{d}^{(r)} + \frac{\alpha^2}{2}(\mathbf{d}^{(r)})^T \nabla^2 f(\mathbf{x}^{(r)}) \mathbf{d}^{(r)} + \frac{\alpha^3 L_2}{6} \|\mathbf{d}^{(r)}\|^3. \tag{18}$$

Then, given by (18) and (61), the minimum amount of the objective decrease, denoted as $d_{\mathbf{v}}$, can be expressed below:

$$d_{\mathbf{v}} \triangleq \alpha q_\pi(\mathbf{x}^{(r)})^T \mathbf{v}(\mathbf{x}^{(r)}) - \left(1 - \frac{\alpha L_2}{3\epsilon'_H(\delta)}\right) \frac{\alpha^2 \epsilon'_H(\delta)}{2}. \tag{19}$$

It can be shown that, if $\mathbf{d}^{(r)}$ is chosen by $\mathbf{v}(\mathbf{x}^{(r)})$, the line search will terminate with $\alpha \ge 9\epsilon'_H(\delta)/(8L_2)$ (see the proof of Lemma 6 for details). Therefore, the minimum decrease in this case is at least

$$d_{\mathbf{v}} = \frac{9\epsilon'_H(\delta)}{8L_2} q_\pi(\mathbf{x}^{(r)})^T \mathbf{v}(\mathbf{x}^{(r)}) - \frac{81}{128} \frac{5}{8} \frac{\epsilon'^3_H(\delta)}{L_2^2}. \tag{20}$$

If $d_q < d_{\mathbf{v}}$, or equivalently

$$\frac{3L_1 \epsilon'_H(\delta)}{L_2} q_\pi(\mathbf{x}^{(r)})^T \mathbf{v}(\mathbf{x}^{(r)}) - \frac{135 L_1 \epsilon'^3_H(\delta)}{128 L_2^2} \ge -\|q_\pi(\mathbf{x}^{(r)})\|^2 \tag{21}$$

where $q_\pi(\mathbf{x}^{(r)})^T \mathbf{v}(\mathbf{x}^{(r)}) \le 0$, it implies that choosing $q_\pi(\mathbf{x}^{(r)})$ may provide more descent of the objective value. □

Next, we will give the detailed algorithm description of the line search used in SNAP.

### A.2  Line Search Algorithm

To understand the algorithm, let us first define the set of *inactive constraints* as

$$\mathbf{A}_{\overline{\mathcal{A}}}(\mathbf{x}) \triangleq [\ \cdots \ \ \mathbf{A}_i \ \ \cdots \ ]^T \in \mathbb{R}^{|\overline{\mathcal{A}}(\mathbf{x})| \times d}, \quad \forall i \in \overline{\mathcal{A}}(\mathbf{x}). \tag{22}$$

The details of the line search algorithm are shown in Algorithm 2. In particular, we will first decide a maximum step-size $\alpha_{\max}^{(r)} > 0$. Recall that $\mathbf{A}_{\overline{\mathcal{A}}}(\mathbf{x})$ defined in (22) represents the set of constraints that are inactive at point $\mathbf{x}$.

**Algorithm 2** Line search algorithm
***

1: **Input:** $\mathbf{x}^{(r)}, \mathbf{d}^{(r)}, \epsilon'_H(\delta), \lambda, \mathbf{A}, \mathbf{b}, \mathcal{F}(\mathbf{x}^{(r)})$
2: **if** $\exists i, (\overline{\mathbf{b}}' - \mathbf{A}_{\overline{\mathcal{A}}}(\mathbf{x}^{(r)})\mathbf{x}^{(r)})_i/(\mathbf{A}_{\overline{\mathcal{A}}}(\mathbf{x}^{(r)})\mathbf{d}^{(r)})_i > 0$ **then**
3: ⠀⠀Compute $\alpha_{\max}^{(r)}$ by
$$\alpha_{\max}^{(r)} \triangleq \max\{\alpha > 0 \mid \mathbf{x}^{(r)} + \alpha\mathbf{d}^{(r)} \in \mathcal{X}\} \tag{23}$$
4: **else**
5: ⠀⠀$\alpha_{\max}^{(r)} = 1/L_1$ if $\mathbf{d}^{(r)} = -q_\pi(\mathbf{x}^{(r)})$ otherwise $\alpha_{\max}^{(r)} = 9\epsilon'_H(\delta)/(4L_2)$
6: **end if**
7: Update $\mathbf{x}^{(r+1)}$ by: $\mathbf{x}^{(r+1)} = \mathbf{x}^{(r)} + \alpha_{\max}^{(r)}\mathbf{d}^{(r)}$
8: **if** $f(\mathbf{x}^{(r)} + \alpha_{\max}^{(r)}\mathbf{d}^{(r)}) < f(\mathbf{x}^{(r)})$ **then**
9: ⠀⠀return $(\mathbf{x}^{(r+1)}, \text{flag} = \Diamond)$
10: **else**
11: ⠀⠀$\alpha \leftarrow \alpha_{\max}^{(r)}$
12: ⠀⠀**if** $\mathbf{d}^{(r)} = -q_\pi(\mathbf{x}^{(r)})$ **then**
$$\rho(\alpha) = -\alpha\|q_\pi(\mathbf{x}^{(r)})\|^2 \tag{24}$$
13: ⠀⠀**else**
$$\rho(\alpha) = -\frac{\alpha^2 \epsilon'_H(\delta)}{4} \tag{25}$$
14: ⠀⠀**end if**
15: ⠀⠀**while**
$$f(\mathbf{x}^{(r)} + \alpha\mathbf{d}^{(r)}) > f(\mathbf{x}^{(r)}) + \frac{1}{2}\rho(\alpha) \tag{26}$$

⠀⠀**do**
16: ⠀⠀⠀$\alpha \leftarrow \frac{1}{2}\alpha$, compute $\rho(\alpha)$.
17: ⠀⠀**end while**
18: ⠀⠀$\alpha^{(r)} \leftarrow \alpha$
19: ⠀⠀$\mathbf{x}^{(r+1)} = \mathbf{x}^{(r)} + \alpha^{(r)}\mathbf{d}^{(r)}$;
20: ⠀⠀return $(\mathbf{x}^{(r+1)}, \text{flag} = \emptyset)$
21: **end if**
***

**Lemma 2.** *If there exists an index* $i \in \overline{\mathcal{A}}(\mathbf{x}^{(r)})$ *such that the following holds*
$$\alpha_i^{(r)} \triangleq \frac{(\mathbf{b}_{\overline{\mathcal{A}}} - \mathbf{A}_{\overline{\mathcal{A}}}(\mathbf{x}^{(r)})\mathbf{x}^{(r)})_i}{(\mathbf{A}_{\overline{\mathcal{A}}}(\mathbf{x}^{(r)})\mathbf{d}^{(r)})_i} > 0, \tag{27}$$

*then, there exists a finite* $\alpha_{\max}^{(r)}$ *given below, which satisfies* (23)
$$\alpha_{\max}^{(r)} = \left\{\min\{\alpha_i^{(r)} > 0\} \mid (\mathbf{A}_{\overline{\mathcal{A}}}(\mathbf{x}^{(r)})\mathbf{x}^{(r)} + \alpha_i^{(r)}\overline{\mathbf{A}}'(\mathbf{x}^{(r)})\mathbf{d}^{(r)})_i = (\mathbf{b}_{\overline{\mathcal{A}}})_i\right\}. \tag{28}$$

*Proof.* Since $\mathbf{x}^{(r)}$ is within the feasible set, then based on the definition of inactive set we have $\mathbf{A}_{\overline{\mathcal{A}}}(\mathbf{x}^{(r)})\mathbf{x}^{(r)} < \mathbf{b}_{\overline{\mathcal{A}}}$. The largest step-size along the direction $\mathbf{d}^{(r)}$ is determined by the largest distance in which the boundary of the feasible solution will be touched, see (23). According to the update rule of the iterate, we need
$$\mathbf{A}_{\overline{\mathcal{A}}}(\mathbf{x}^{(r)})(\mathbf{x}^{(r)} + \alpha\mathbf{d}^{(r)}) \leq \mathbf{b}_{\overline{\mathcal{A}}}, \tag{29}$$
which is equivalent to the following component-wise form:
$$\alpha\left(\mathbf{A}_{\overline{\mathcal{A}}}(\mathbf{x}^{(r)})\mathbf{d}^{(r)}\right)_i \leq \left(\mathbf{b}_{\overline{\mathcal{A}}} - \mathbf{A}_{\overline{\mathcal{A}}}(\mathbf{x}^{(r)})\mathbf{x}^{(r)}\right)_i, \quad \forall i. \tag{30}$$

If $(\mathbf{A}_{\overline{\mathcal{A}}}(\mathbf{x}^{(r)})\mathbf{d}^{(r)})_i \leq 0$, then any $\alpha > 0$ can satisfy (30). Alternatively, suppose $(\mathbf{A}_{\overline{\mathcal{A}}}(\mathbf{x}^{(r)})\mathbf{d}^{(r)})_i > 0$. Note that due to the feasibility of $\mathbf{x}^{(r)}$, and the definition of inactive set, we have $(\mathbf{b}_{\overline{\mathcal{A}}} - \mathbf{A}_{\overline{\mathcal{A}}}(\mathbf{x}^{(r)})\mathbf{x}^{(r)})_i > 0$. Then it is possible to pick a positive $\alpha$ satisfying $\alpha = (\mathbf{b}_{\overline{\mathcal{A}}} - \mathbf{A}_{\overline{\mathcal{A}}}(\mathbf{x}^{(r)})\mathbf{x}^{(r)})_i/(\mathbf{A}_{\overline{\mathcal{A}}}(\mathbf{x}^{(r)})\mathbf{d}^{(r)})_i$. That is, going along the current direction far enough will eventually reach the boundary of the feasible set.

Then it follows that if there exists a *finite* step-size $\alpha$ so that
$$(\mathbf{A}_{\overline{\mathcal{A}}}(\mathbf{x}^{(r)})(\mathbf{x}^{(r)} + \alpha\mathbf{d}^{(r)}))_i = (\mathbf{b}_{\overline{\mathcal{A}}})_i, \tag{31}$$

we can easily compute $\alpha_{\max}^{(r)}$ in the closed-form by (28). $\qquad\square$

On the other hand, if the condition (27) does not hold, it means that along the current direction the problem is effectively *unconstrained*. Therefore, the line search algorithm reduces to the classic unconstrained update. Then by setting $\alpha_{\max}^{(r)} = 1/L_1$, SNAP will give a sufficient decrease in this case; see Lemma 5.

After choosing $\alpha_{\max}^{(r)}$, we check if the following holds: $f(\mathbf{x}^{(r)} + \alpha_{\max}^{(r)} \mathbf{d}^{(r)}) < f(\mathbf{x}^{(r)})$. If so, then the algorithm either touches the boundary without increasing the objective, or it has already achieved sufficient descent.

If the objective increases, then the algorithm will call the backtracking line search by successively shrinking the step-size starting at $\alpha \leftarrow \alpha_{\max}^{(r)}$. In particular, if $f(\mathbf{x}^{(r)} + \alpha \mathbf{d}^{(r)}) > f(\mathbf{x}^{(r)}) + \lambda \rho(\alpha)$ (where $\rho(\alpha)$ is some pre-determined negative quantity, see (24)–(25)), we will implement $\alpha \leftarrow \frac{1}{2}\alpha$ until a sufficient descent is satisfied (note, such a sufficient descent can be eventually achieved, see Lemma 5 and Lemma 6).

# B    Proofs Related to Stationary Points

## B.1    Proof of Proposition 1

*Proof.* When $f(\mathbf{x}) = g(\mathbf{x}) + \mathbf{q}^T \mathbf{x}$, the KKT conditions of (1) are given by

$$\nabla g(\mathbf{x}^*) + \sum_{j \in \mathcal{A}(\mathbf{x}^*)} \boldsymbol{\mu}_j^* \mathbf{A}_j = -\mathbf{q}, \tag{32a}$$

$$\mathbf{A}_j \mathbf{x}^* = \mathbf{b}_j, \quad \boldsymbol{\mu}_j \geq 0 \quad \forall j \in \mathcal{A}(\mathbf{x}^*), \tag{32b}$$

$$\mathbf{A}_j \mathbf{x}^* < \mathbf{b}_j, \quad \boldsymbol{\mu}_j^* = 0, \quad \forall j \notin \mathcal{A}(\mathbf{x}^*). \tag{32c}$$

If there is no active constraint at point $\mathbf{x}^*$, then SC condition holds automatically. Here we assume that $\mathbf{x}^*$ has at least one active constraint, we have $|\mathcal{A}(\mathbf{x}^*)| \geq 1$.

We prove the claim by contradiction. Assume that the strict complementarity condition does not hold at $\mathbf{x}^*$. Without loss of generality, assume that $\mathbf{A}_1 \mathbf{x}^* = \mathbf{b}_1$ and $\boldsymbol{\mu}_1^* = 0$. Consider the Lipschitz continuous map $\Phi$ defined below

$$\Phi(\mathbf{x}^*, \boldsymbol{\mu}^*) = \nabla g(\mathbf{x}^*) + \sum_{j \in \mathcal{A}(\mathbf{x}^*)} \boldsymbol{\mu}_j^* \mathbf{A}_j. \tag{33}$$

This is a mapping from the set $\mathcal{T}$ to the entire space of $\mathbb{R}^d$ (because $\mathbf{q}$ is generated from a continuous measure in $\mathbb{R}^d$), where the set $\mathcal{T}$ is given below:

$$\mathcal{T} = \{(\mathbf{x}^*, \boldsymbol{\mu}^*) | \mathbf{A}_j \mathbf{x}^* = \mathbf{b}_j, \boldsymbol{\mu}_j^* \geq 0, j \in \mathcal{A}(\mathbf{x}^*), \mathbf{A}_{j'} \mathbf{x}^* < \mathbf{b}_{j'}, \boldsymbol{\mu}_{j'}^* = 0, j' \notin \mathcal{A}(\mathbf{x}^*)\}. \tag{34}$$

In the following, we will quantify the dimension of $\mathcal{T}$. By assumption, all the $\mathbf{A}_j$'s with $j \in \mathcal{A}(\mathbf{x}^*)$ are linearly independent, that is, $\mathbf{A}_{\mathcal{A}}(\mathbf{x}^*)$ is a full row rank matrix. It follows that $\mathbf{b}_{\mathcal{A}}$ is in the range space of matrix $\mathbf{A}_{\mathcal{A}}(\mathbf{x}^*)$. Since $\mathbf{A}_{\mathcal{A}}(\mathbf{x}^*)\mathbf{x}^* = \mathbf{b}_{\mathcal{A}}$, we know that the dimension of the active space of $\mathbf{x}^*$ is the rank of $\mathbf{A}_{\mathcal{A}}(\mathbf{x}^*)$, meaning that the dimension of the free space of $\mathbf{x}^*$ is the rank of $\mathsf{Null}(\mathbf{A}_{\mathcal{A}}(\mathbf{x}^*))$, i.e., $(d - |\mathcal{A}(\mathbf{x}^*)|)$. [2] Note that there are $|\mathcal{A}(\mathbf{x}^*)|$ active constraints and $\boldsymbol{\mu}_1^* = 0$, so the dimension of the free space of vector $\boldsymbol{\mu}^*$ is $(|\mathcal{A}(\mathbf{x}^*)| - 1)$. Therefore, $\Phi$ maps from a $(d - 1)$-dimensional subspace to a $d$-dimensional space, implying that the image of the mapping is zero-measure in $\mathbb{R}^d$. However, $\mathbf{q}$ is generated from a continuous measure, which results in a contradiction of the assumption that the strict complementarity condition does not hold. $\square$

## B.2    Proof of Corollary 1

*Proof.* We apply the same proof technique in Proposition 1 to show the claim of Corollary 1. Let $\mathcal{S}(\mathbf{x}^*) \triangleq \{j \mid \mathbf{A}_j, \forall j \in \mathcal{A}(\mathbf{x}^*) \text{ are linearly independent}\}$ and let $\bar{\mathcal{S}}(\mathbf{x}^*)$ denote the complement of set $\mathcal{S}(\mathbf{x}^*)$. Clearly, $\mathcal{S}(\mathbf{x}^*)$ is a subset of $\mathcal{A}(\mathbf{x}^*)$. First, we define the matrix $\mathbf{A}_{\mathcal{S}}(\mathbf{x}^*)$ as

$$\mathbf{A}_{\mathcal{S}}(\mathbf{x}^*) \triangleq \begin{bmatrix} \vdots \\ \mathbf{A}_j \\ \vdots \end{bmatrix} \in \mathbb{R}^{|\mathcal{S}(\mathbf{x}^*)| \times d}, \quad \forall j \in \mathcal{S}(\mathbf{x}^*). \tag{35}$$

Obviously, $\mathbf{A}_{\mathcal{S}}(\mathbf{x}^*)$ is a full row rank matrix, where the rank of $\mathbf{A}_{\mathcal{S}}(\mathbf{x}^*)$ is the size of $\mathcal{S}(\mathbf{x}^*)$, i.e., $|\mathcal{S}(\mathbf{x}^*)|$. In the following, we will show that the number of simultaneously active constraints is at most $|\mathcal{S}(\mathbf{x}^*)|$. We prove the claim by contradiction. Consider $i \in \overline{\mathcal{S}}(\mathbf{x}^*)$. Since $i \notin \mathcal{S}(\mathbf{x}^*)$, $\mathbf{A}_i$ can be linearly represented by $\mathbf{A}_j$s $j \in \mathcal{A}(\mathbf{x}^*)$, i.e.,

$$\mathbf{A}_i = \sum_j \alpha_j \mathbf{A}_j, j \in \mathcal{A}(\mathbf{x}^*), \tag{36}$$

where there exists at least one $\alpha_j$ which is not zero. Since $i, j \in \mathcal{A}(\mathbf{x}^*)$, we have $\mathbf{A}_j \mathbf{x}^* = \mathbf{b}_j$. Combining (36), we have $\sum_j \alpha_j \mathbf{b}_j = \mathbf{b}_i$. Since $\mathbf{b}_i$ is generated from a continuous measure, $\sum_j \alpha_j \mathbf{b}_j = \mathbf{b}_i$ will not hold with high probability. We have a contradiction. Therefore, we can conclude that the dimension of the free space of $\mathbf{x}^*$ is at least $d - |\mathcal{S}(\mathbf{x}^*)|$.

Next, we use the same argument as the proof of Proposition 1 to quantify the dimension of $\boldsymbol{\mu}^*$. Since there are $|\mathcal{S}(\mathbf{x}^*)|$ active constraints and $\boldsymbol{\mu}_1^* = 0$, the dimension of $\boldsymbol{\mu}^*$ is at most $|\mathcal{S}(\mathbf{x}^*)| - 1$. Thus, the dimension of $\mathcal{T}$ is $d - 1$, meaning that $\Phi$ defined in (33) maps from a $d - 1$ dimension subset to a $d$-dimensional space. Therefore, the image is zero-measure in $\mathbb{R}^d$. However, $\mathbf{q}$ is generated from a continuous measure, which again results in a contradiction of the assumption that the strict complementarity condition does not hold. $\qquad\square$

## B.3 Equivalence of First-Order Conditions

**Lemma 3.** *The conditions* (8a) *and* (10a) *are equivalent. That is, for any* $(\mathbf{x}^*, \widetilde{\epsilon}_G)$ *that satisfies* (10a)*, it also satisfies* (8a)*. Alternatively, for any* $(\mathbf{x}^*, \epsilon_G)$ *that satisfies* (8a)*, then* $(\mathbf{x}^*, \widetilde{\epsilon}_G)$ *satisfies* (10a)*, with* $\widetilde{\epsilon}_G \triangleq \epsilon_G \left( \alpha \|\nabla f(\mathbf{x}^*)\| + 1 + \alpha \epsilon_G \right)$.

*Proof.* First, suppose $\mathbf{x}^*$ and $\widetilde{\epsilon}_G$ together satisfy (10a). Let us define $\widetilde{\mathbf{x}} \triangleq \pi_{\mathcal{X}}(\mathbf{x}^* - \alpha \nabla f(\mathbf{x}^*))$, that is, from the definition of $\pi_{\mathcal{X}}(\mathbf{v})$ (i.e., $\pi_{\mathcal{X}}(\mathbf{v}) \triangleq \arg\min_{\mathbf{w} \in \mathcal{X}} \|\mathbf{w} - \mathbf{v}\|^2$) we have

$$\widetilde{\mathbf{x}} = \arg\min_{\mathbf{y} \in \mathcal{X}} \|\mathbf{x}^* - \alpha \nabla f(\mathbf{x}^*) - \mathbf{y}\|^2. \tag{37}$$

From the optimality condition of (37), we know

$$\langle \widetilde{\mathbf{x}} - (\mathbf{x}^* - \alpha \nabla f(\mathbf{x}^*)), \mathbf{y} - \widetilde{\mathbf{x}} \rangle \geq 0, \ \forall \mathbf{y} \in \mathcal{X}. \tag{38}$$

Substituting $\mathbf{y} = \mathbf{x}^*$ into (38), we have $\langle \widetilde{\mathbf{x}} - (\mathbf{x}^* - \nabla f(\mathbf{x}^*)), \mathbf{x}^* - \widetilde{\mathbf{x}} \rangle \geq 0$, which implies $\alpha \langle \nabla f(\mathbf{x}^*), \widetilde{\mathbf{x}} - \mathbf{x}^* \rangle \leq -\|\widetilde{\mathbf{x}} - \mathbf{x}^*\|^2$. Therefore, we have

$$\|\widetilde{\mathbf{x}} - \mathbf{x}^*\| \leq -\alpha \left\langle \nabla f(\mathbf{x}^*), \frac{\widetilde{\mathbf{x}} - \mathbf{x}^*}{\|\widetilde{\mathbf{x}} - \mathbf{x}^*\|} \right\rangle \overset{(10a)}{\leq} \widetilde{\epsilon}_G,$$

meaning that $\|g_\pi(\mathbf{x}^*)\| \leq \widetilde{\epsilon}_G$. This direction is completed.

Second, we suppose $\mathbf{x}^* \in \mathcal{X}$ and $\epsilon_G > 0$ together satisfy (8a). Again let us define $\widetilde{\mathbf{x}} \triangleq \pi_{\mathcal{X}}(\mathbf{x}^* - \alpha \nabla f(\mathbf{x}^*))$. Consider an arbitrary point $\mathbf{y} \in \mathcal{X}$ and $\widetilde{\mathbf{x}} + \theta(\mathbf{y} - \widetilde{\mathbf{x}}) \in \mathcal{X}$ where $\theta \in (0, 1)$. We have, for all $\mathbf{y} \in \mathcal{X}$, the following holds:

$$\|\mathbf{x}^* - \alpha \nabla f(\mathbf{x}^*) - \widetilde{\mathbf{x}}\|^2 \leq \|\mathbf{x}^* - \alpha \nabla f(\mathbf{x}^*) - (\widetilde{\mathbf{x}} + \theta(\mathbf{y} - \widetilde{\mathbf{x}}))\|^2$$
$$= \|\mathbf{x}^* - \alpha \nabla f(\mathbf{x}^*) - \widetilde{\mathbf{x}}\|^2 - 2\theta \langle \mathbf{x}^* - \alpha \nabla f(\mathbf{x}^*) - \widetilde{\mathbf{x}}, \mathbf{y} - \widetilde{\mathbf{x}} \rangle + \theta^2 \|\mathbf{y} - \widetilde{\mathbf{x}}\|^2,$$

which is equivalent to

$$\langle \mathbf{x}^* - \alpha \nabla f(\mathbf{x}^*) - \widetilde{\mathbf{x}}, \mathbf{y} - \widetilde{\mathbf{x}} \rangle \leq \frac{\theta}{2} \|\mathbf{y} - \widetilde{\mathbf{x}}\|^2, \ \ \forall \mathbf{y} \in \mathcal{X}. \tag{39}$$

The right-hand-side (RHS) of (39) can be made arbitrarily small by $\theta$ for a given $\mathbf{y}$, so LHS of (39) cannot be strictly positive. This further implies that

$$\langle \nabla f(\mathbf{x}^*), \mathbf{y} - \mathbf{x}^* \rangle \geq \frac{1}{\alpha} \langle \mathbf{x}^* - \widetilde{\mathbf{x}}, \mathbf{y} - \widetilde{\mathbf{x}} \rangle + \langle \nabla f(\mathbf{x}^*), \widetilde{\mathbf{x}} - \mathbf{x}^* \rangle$$

$$\overset{(a)}{\geq} - \epsilon_G (\|\mathbf{y} - \widetilde{\mathbf{x}}\| + \alpha \|\nabla f(\mathbf{x}^*)\|) \tag{40}$$

$$\geq - \epsilon_G (\|\mathbf{y} - \mathbf{x}^*\| + \|\mathbf{x}^* - \widetilde{\mathbf{x}}\| + \alpha \|\nabla f(\mathbf{x}^*)\|) \tag{41}$$

$$\overset{(b)}{\geq} - \epsilon_G (\alpha \|\nabla f(\mathbf{x}^*)\| + 1 + \alpha \epsilon_G) \triangleq -\widetilde{\epsilon}_G \tag{42}$$

where in $(a)$ we use $(1/\alpha)\|\mathbf{x}^* - \widetilde{\mathbf{x}}\| \leq \epsilon_G$ and Cauchy-Schwartz inequality, in $(b)$ we know $\|\mathbf{y} - \mathbf{x}^*\| \leq 1$ from condition (10a). $\qquad\square$

## B.4 Proof of Proposition 2

*Proof.* The equivalence between FOSP1 and FOSP2 has been shown in Lemma 3. Below, we focus on the equivalence of the second-order optimality conditions.

**Sufficiency.** First, assume that (8b) holds. For a given $\mathbf{x}$ satisfying $\langle \nabla f(\mathbf{x}^*), \mathbf{x} - \mathbf{x}^* \rangle = 0$, by applying (11a), we have $\sum_{j=1}^{m} \boldsymbol{\mu}_j^* \langle \mathbf{A}_j, \mathbf{x} - \mathbf{x}^* \rangle = 0$. Further by (3), we can decompose the previous sum into the following:

$$\sum_{j \in \mathcal{A}(\mathbf{x}^*)} \boldsymbol{\mu}_j^* \langle \mathbf{A}_j, \mathbf{x} - \mathbf{x}^* \rangle + \sum_{j \notin \mathcal{A}(\mathbf{x}^*)} \boldsymbol{\mu}_j^* \langle \mathbf{A}_j, \mathbf{x} - \mathbf{x}^* \rangle = 0. \tag{43}$$

Combining (43) with the complementarity conditions in (11b), we have

$$\sum_{j \in \mathcal{A}(\mathbf{x}^*)} \boldsymbol{\mu}_j^* \langle \mathbf{A}_j, \mathbf{x} - \mathbf{x}^* \rangle = 0. \tag{44}$$

Also note that for each $\mathbf{x} \in \mathcal{X}$, and each active constraint $j \in \mathcal{A}(\mathbf{x}^*)$, we have $\mathbf{A}_j \mathbf{x} \leq \mathbf{b}_j = \mathbf{A}_j \mathbf{x}^*, \quad \forall j \in \mathcal{A}(\mathbf{x}^*)$. It follows that $\langle \mathbf{A}_j, \mathbf{x} - \mathbf{x}^* \rangle \leq 0, \forall j \in \mathcal{A}(\mathbf{x}^*)$, and $\forall \mathbf{x} \in \mathcal{X}$. Due to the assumed strict complementarity condition, we have $\boldsymbol{\mu}_j^* > 0, j \in \mathcal{A}(\mathbf{x}^*)$.

Combining the above two facts, we conclude that each term in (44) is nonpositive. However, the requirement that the sum of them equals to zero implies that $\langle \mathbf{A}_j, \mathbf{x} - \mathbf{x}^* \rangle = 0, \forall j \in \mathcal{A}(\mathbf{x}^*)$. From (8b), we know that $\forall \mathbf{y}, \mathbf{A}_{\mathcal{A}}(\mathbf{x}^*)\mathbf{y} = 0$, we have $\mathbf{y}^T \nabla^2 f(\mathbf{x}^*)\mathbf{y} \geq 0$.

**Necessity.** Second, let us suppose that $\mathbf{x}^*$ satisfies the exact first-order stationary point, and for each *feasible* $\mathbf{x} \in \mathcal{X}$ that satisfies $\langle \nabla f(\mathbf{x}^*), \mathbf{x} - \mathbf{x}^* \rangle = 0$, we have

$$(\mathbf{x} - \mathbf{x}^*)^T \nabla^2 f(\mathbf{x}^*)(\mathbf{x} - \mathbf{x}^*) \geq 0. \tag{45}$$

Suppose at the KKT point $\mathbf{x}^*, \boldsymbol{\mu}^*$ the SC is satisfied. Further we assume that for the inactive set, the following holds:

$$\mathbf{A}_i \mathbf{x}^* + \epsilon_i = \mathbf{b}_i, \text{ for some } \epsilon_i > 0, \quad \forall i \in \bar{\mathcal{A}}(\mathbf{x}^*).$$

Let $\epsilon = \min_i \{\epsilon_i\} > 0$. Take the inner product between $\mathbf{x} - \mathbf{x}^*$ and left-hand-side (LHS) of (11a) we have

$$\langle \nabla f(\mathbf{x}^*), \mathbf{x} - \mathbf{x}^* \rangle = -\sum_{j=1}^{m} \boldsymbol{\mu}_j^* \langle \mathbf{A}_j, \mathbf{x} - \mathbf{x}^* \rangle \overset{(11b)}{=} -\sum_{j \in \mathcal{A}(\mathbf{x}^*)} \boldsymbol{\mu}_j^* \langle \mathbf{A}_j, \mathbf{x} - \mathbf{x}^* \rangle. \tag{46}$$

By SC condition we have $\mathbf{A}_j(\mathbf{x} - \mathbf{x}^*) = 0, \forall j \in \mathcal{A}(\mathbf{x}^*)$. Consider any $\mathbf{y}$ that satisfies

$$\mathbf{A}_j \mathbf{y} = 0, \quad \forall j \in \mathcal{A}(\mathbf{x}^*). \tag{47}$$

First, we argue that, if the following holds, then there exists $\mathbf{x} \in \mathcal{X}$ such that $\mathbf{y} = \mathbf{x} - \mathbf{x}^*$:

$$\mathbf{A}_j \mathbf{y} = 0, \forall j \in \mathcal{A}(\mathbf{x}^*), \quad \mathbf{A}_i \mathbf{y} \leq \frac{\epsilon}{2}, \forall i \in \bar{\mathcal{A}}(\mathbf{x}^*). \tag{48}$$

By setting $\mathbf{y} = \mathbf{z} - \mathbf{x}^*$, for any $\mathbf{z}$, we obtain $\mathbf{A}_i(\mathbf{z} - \mathbf{x}^*) \leq \epsilon/2$, which implies $\mathbf{A}_i \mathbf{z} \leq \mathbf{A}_i \mathbf{x}^* + \frac{\epsilon}{2} < \mathbf{b}_i$ where the last inequality is due to the definition of $\epsilon$. Further, for the active set, it is clear that $\mathbf{A}_j \mathbf{z} = \mathbf{A}_j \mathbf{x}^* = \mathbf{b}_j, \forall j \in \mathcal{A}(\mathbf{x}^*)$, so $\mathbf{z}$ is feasible. Suppose that for a given $\mathbf{y}$ satisfying (47), we cannot find any $\mathbf{x} \in \mathcal{X}$ such that $\mathbf{y} = \mathbf{x} - \mathbf{x}^*$, then it must be the case that there exists a subset $\mathcal{Q} \in \bar{\mathcal{A}}(\mathbf{x}^*)$ such that $\mathbf{A}_q \mathbf{y} = \theta_q > \frac{\epsilon}{2}, \forall q \in \mathcal{Q}$. Let us define $\theta_{\max} := \max_q \{\theta_q\}$, and $\widetilde{\mathbf{y}} = \frac{1}{\theta_{\max}} \frac{\epsilon}{2} \mathbf{y}$, and note that $\frac{1}{\theta_{\max}} \frac{\epsilon}{2} < 1$. Then for this new $\widetilde{\mathbf{y}}$, the following holds

$$\mathbf{A}_j \widetilde{\mathbf{y}} = 0, \forall j \in \mathcal{A}(\mathbf{x}^*), \qquad \mathbf{A}_q \widetilde{\mathbf{y}} \leq \epsilon/2, \forall q \in \mathcal{Q}$$

$$\mathbf{A}_j \widetilde{\mathbf{y}} = \frac{1}{\theta_{\max}} \frac{\epsilon}{2} \mathbf{A}_j \mathbf{y}, \forall j \in \bar{\mathcal{A}}(\mathbf{x}^*), j \notin \mathcal{Q}.$$

Note that for all $j \in \bar{\mathcal{A}}(\mathbf{x}^*), j \notin \mathcal{Q}, \mathbf{A}_j \mathbf{y} \leq \epsilon/2$. We have the following two cases for those indices.

**Case 1.** First, if $\mathbf{A}_j \mathbf{y} \leq 0$ then it is clear that

$$\mathbf{A}_j \widetilde{\mathbf{y}} = \frac{1}{\theta_{\max}} \frac{\epsilon}{2} \mathbf{A}_j \mathbf{y} \leq 0.$$

**Case 2.** Second, if $0 \le \mathbf{A}_j\mathbf{y} \le \epsilon/2$ then it is clear that

$$\mathbf{A}_j\widetilde{\mathbf{y}} = \frac{1}{\theta_{\max}}\frac{\epsilon}{2}\mathbf{A}_j\mathbf{y} \overset{(a)}{\le} \mathbf{A}_j\mathbf{y} \le \epsilon/2$$

where $(a)$ uses the fact that $\frac{1}{\theta_{\max}}\frac{\epsilon}{2} < 1$, and $0 \le \mathbf{A}_j\mathbf{y}$.

Overall we have $\mathbf{A}_i\widetilde{\mathbf{y}} \le \epsilon/2, \forall\, i \in \bar{\mathcal{A}}(\mathbf{x}^*)$, and $\mathbf{A}_j\widetilde{\mathbf{y}} = 0, \forall\, i \in \mathcal{A}(\mathbf{x}^*)$, i.e., condition (48) holds for $\widetilde{\mathbf{y}}$. Therefore, there must exist $\mathbf{x} \in \mathcal{X}$ such that $\widetilde{\mathbf{y}} = \mathbf{x} - \mathbf{x}^*$.

We conclude that for any $\mathbf{y}$ satisfying $\mathbf{A}_i\mathbf{y} = 0, \forall\, i \in \mathcal{A}(\mathbf{x}^*)$, there exists a constant $\theta$ and $\mathbf{x} \in \mathcal{X}$ such that $\theta\mathbf{y} = \mathbf{x} - \mathbf{x}^*$. By (45), we obtain

$$\theta^2\mathbf{y}^T\nabla^2 f(\mathbf{x}^*)\mathbf{y} \ge 0, \quad \text{or equivalently,} \quad \mathbf{y}^T\nabla^2 f(\mathbf{x}^*)\mathbf{y} \ge 0. \tag{49}$$

This direction is proved. $\qquad\square$

### B.5 Proof of Corollary 2

*Proof.* It can be easily checked that Corollary 2 is true from the proof of Proposition 2 by letting $\mathbf{y} = \mathbf{x} - \mathbf{x}^*$ and considering $\|\mathbf{y}\| \le 1$. $\qquad\square$

### B.6 Proof of Proposition 3

*Proof.* Let $\mathbf{x}^*$ be a limit point of the sequence $\{\mathbf{x}^{(r)}\}$. By restricting to a subsequence if necessary, let us assume that $\lim_{r\to\infty}\mathbf{x}^{(r)} = \mathbf{x}^*$. First notice that the function $g_\pi(\cdot)$ is continuous based on its definition. Therefore, $g_\pi(\mathbf{x}^*) = \lim_{r\to\infty} g_\pi(\mathbf{x}^r) = 0$. Therefore, (7a) is satisfied at the point $\mathbf{x}^*$.

In order to show (7b), let us define $\mathcal{Y}^{(r)} \triangleq \{\mathbf{y} \mid \mathbf{A}_{\mathcal{A}}(\mathbf{x}^{(r)})\mathbf{y} = 0\}$ and $\mathcal{Y}^* \triangleq \{\mathbf{y} \mid \mathbf{A}_{\mathcal{A}}(\mathbf{x}^*)\mathbf{y} = 0\}$. We first prove that there exists an index $r'$ such that $\mathcal{Y}^* \subseteq \mathcal{Y}^{(r)}, \forall r \ge r'$. To show that, first consider an inactive index $j \in \bar{\mathcal{A}}(\mathbf{x}^*)$. Clearly, $\mathbf{A}_j\mathbf{x}^* \ne \mathbf{b}$ and therefore, there exists an index $r'_j$ such that $\mathbf{A}_j\mathbf{x}^{(r)} \ne \mathbf{b}, \forall r \ge r'_j$. Thus, $j \in \bar{\mathcal{A}}(\mathbf{x}^{(r)}), \forall r \ge r'_j$. By repeating this argument for all indices $j$ and setting $r' = \max_j\{r_j\}$, we have $\bar{\mathcal{A}}(\mathbf{x}^*) \subseteq \bar{\mathcal{A}}(\mathbf{x}^{(r)}), \forall r \ge r'$. Therefore, $\mathcal{A}(\mathbf{x}^{(r)}) \subseteq \mathcal{A}(\mathbf{x}^*), \forall r \ge r'$, which immediately implies that

$$\mathcal{Y}^* \subseteq \mathcal{Y}^{(r)}, \forall r \ge r'. \tag{50}$$

Furthermore, using the definition of exact SOSP1, we have

$$-\epsilon_H^{(r)} \le \min_{\mathbf{y}} \quad \mathbf{y}^T\nabla^2 f(\mathbf{x}^{(r)})\mathbf{y}, \quad \text{s.t.} \quad \mathbf{y} \in \mathcal{Y}^{(r)}.$$

By letting $r \to \infty$ and using (50), we obtain

$$0 \le \min_{\mathbf{y}} \quad \mathbf{y}^T\nabla^2 f(\mathbf{x}^*)\mathbf{y}, \quad \text{s.t.} \quad \mathbf{y} \in \mathcal{Y}^*.$$

Therefore, (7b) is satisfied at the point $\mathbf{x}^*$. $\qquad\square$

## C  Proofs of SNAP

In this section, we show that SNAP converges to an $(\epsilon_G, \epsilon_H)$-SOSP1 in a finite number of steps. In Algorithm 2, it can be observed that $\mathbf{d}^{(r)}$ could be chosen by projected gradient $q_\pi(\mathbf{x}^{(r)})$ or negative curvature $\mathbf{v}(\mathbf{x}^{(r)})$. Using the line search algorithm ensures that the iterates stay in the feasible set. When $\alpha_{\max}^{(r)}$ is chosen by (23), the objective function will not increase. When $\alpha_{\max}^{(r)}$ is not chosen by (23), we will have a sufficient descent. We will give the following three lemmas that quantify the minimum decrease of the objective value by implementing one step of the algorithm, i.e., $\mathbf{x}^{(r+1)} = \mathbf{x}^{(r)} + \alpha^{(r)}\mathbf{d}^{(r)}$. They serve as the stepping stones for the main result that follows.

The descent Lemma of PGD is given by the following.

### C.1 Descent Lemmas

**Lemma 4.** *If $\mathbf{x}^{(r+1)}$ is computed by projected gradient descent with step-size chosen by $1/L_1$, then $f(\mathbf{x}^{(r+1)}) \leq f(\mathbf{x}^{(r)}) - \frac{\epsilon_G^2}{18L_1}$.*

*Proof.* The proof follows the classic theory of the projected gradient descent. According to the optimality condition of the projection, we have

$$\left\langle \mathbf{x}^{(r+1)} - (\mathbf{x}^{(r)} - \alpha_\pi \nabla f(\mathbf{x}^{(r)})), \mathbf{x} - \mathbf{x}^{(r+1)} \right\rangle \geq 0 \quad \mathbf{x} \in \mathcal{X}. \tag{51}$$

Applying this relation with $\mathbf{x} = \mathbf{x}^{(r)}$, we obtain

$$\left\langle \nabla f(\mathbf{x}^{(r)}), \mathbf{x}^{(r+1)} - \mathbf{x}^{(r)} \right\rangle \leq -\frac{1}{\alpha_\pi} \|\mathbf{x}^{(r+1)} - \mathbf{x}^{(r)}\|^2. \tag{52}$$

According to $L_1$-Lipschitz continuity, we have

$$f(\mathbf{x}^{(r+1)}) - f(\mathbf{x}^{(r)}) \leq \nabla f(\mathbf{x}^{(r)})^T (\mathbf{x}^{(r+1)} - \mathbf{x}^{(r)}) + \frac{L_1}{2} \|\mathbf{x}^{(r+1)} - \mathbf{x}^{(r)}\|^2, \tag{53}$$

where

$$\mathbf{x}^{(r+1)} = \pi_\mathcal{X}(\mathbf{x}^{(r)} - \alpha_\pi \nabla f(\mathbf{x}^{(r)})). \tag{54}$$

Then, we have

$$f(\mathbf{x}^{(r+1)}) \leq f(\mathbf{x}^{(r)}) + \left( \frac{L_1}{2} - \frac{1}{\alpha_\pi} \right) \|\mathbf{x}^{(r+1)} - \mathbf{x}^{(r)}\|^2, \tag{55}$$

where $0 < \alpha_\pi \leq 1/L_1$, implying

$$\begin{aligned} f(\mathbf{x}^{(r+1)}) &\leq f(\mathbf{x}^{(r)}) - \frac{L_1}{2} \|\mathbf{x}^{(r+1)} - \mathbf{x}^{(r)}\|^2 \\ &\stackrel{(a)}{\leq} f(\mathbf{x}^{(r)}) - \frac{L_1}{2(\frac{2}{\alpha_\pi} + L_1)^2} \|g_\pi(\mathbf{x}^{(r)})\|^2 \\ &\stackrel{(b)}{\leq} f(\mathbf{x}^{(r)}) - \frac{\epsilon_G^2}{18L_1} \end{aligned}$$

where in $(a)$ we use the nonexpansiveness of the projection operator, and the details are as follows:

$$\begin{aligned} \|g_\pi(\mathbf{x}^{(r)})\| &= \frac{1}{\alpha_\pi} \|\pi_\mathcal{X}(\mathbf{x}^{(r)} - \alpha_\pi \nabla f(\mathbf{x}^{(r)})) - \mathbf{x}^{(r)}\| \\ &= \frac{1}{\alpha_\pi} \|\pi_\mathcal{X}(\mathbf{x}^{(r)} - \alpha_\pi \nabla f(\mathbf{x}^{(r)})) - \mathbf{x}^{(r+1)} + \mathbf{x}^{(r+1)} - \mathbf{x}^{(r)}\| \\ &\leq \frac{1}{\alpha_\pi} \|\mathbf{x}^{(r+1)} - \mathbf{x}^{(r)}\| + \frac{1}{\alpha_\pi} \|\mathbf{x}^{(r+1)} - \pi_\mathcal{X}(\mathbf{x}^{(r)} - \alpha_\pi \nabla f(\mathbf{x}^{(r)}))\| \\ &= \frac{1}{\alpha_\pi} \|\mathbf{x}^{(r+1)} - \mathbf{x}^{(r)}\| + \frac{1}{\alpha_\pi} \|\pi_\mathcal{X}(\mathbf{x}^{(r+1)} - \alpha_\pi \nabla f(\mathbf{x}^{(r+1)})) - \pi_\mathcal{X}(\mathbf{x}^{(r)} - \alpha_\pi \nabla f(\mathbf{x}^{(r)}))\| \\ &\leq \frac{2}{\alpha_\pi} \|\mathbf{x}^{(r+1)} - \mathbf{x}^{(r)}\| + \|\nabla f(\mathbf{x}^{(r+1)}) - \nabla f(\mathbf{x}^{(r)})\| \\ &\leq \left( \frac{2}{\alpha_\pi} + L_1 \right) \|\mathbf{x}^{(r+1)} - \mathbf{x}^{(r)}\|; \end{aligned} \tag{56}$$

in $(b)$ we take $\alpha_\pi = 1/L_1$.

From (56), we have the sufficient descent of the objective value if the constant step-size is used. $\quad\square$

**Lemma 5.** *Suppose $\mathbf{d}^{(r)}$ is chosen as $-q_\pi(\mathbf{x}^{(r)})$ and $\mathbf{x}^{(r+1)}$ is computed by the NCD step of Algorithm 1, and line 7 of Algorithm 2 does not hold, i.e., $f(\mathbf{x}^{(r)} + \alpha_{\max}^{(r)} \mathbf{d}^{(r)}) \geq f(\mathbf{x}^{(r)})$. Then, $\alpha_{\max}^{(r)} \geq 1/L_1$ and the line search algorithm terminates with $\alpha \geq 1/(2L_1)$ and a descent of the following can be achieved $f(\mathbf{x}^{(r+1)}) \leq f(\mathbf{x}^{(r)}) - 3\epsilon_H'^3(\delta)/(8L_2^2)$.*

*Proof.* First, according to the $L_1$-Lipschitz continuity, we have

$$
\begin{aligned}
f(\mathbf{x}^{(r+1)}) =& f\left(\mathbf{x}^{(r)} - \alpha q_\pi(\mathbf{x}^{(r)})\right) \\
&\overset{(a)}{\leq} f(\mathbf{x}^{(r)}) - \alpha \nabla f(\mathbf{x}^{(r)})^T q_\pi(\mathbf{x}^{(r)}) + \frac{\alpha^2}{2} L_1 \|q_\pi(\mathbf{x}^{(r)})\|^2 \\
&\overset{(b)}{=} f(\mathbf{x}^{(r)}) - \alpha q_\pi(\mathbf{x}^{(r)})^T q_\pi(\mathbf{x}^{(r)}) + \frac{\alpha^2}{2} L_1 \|q_\pi(\mathbf{x}^{(r)})\|^2 \\
=& f(\mathbf{x}^{(r)}) - \left(\alpha - \frac{\alpha^2}{2} L_1\right) \|q_\pi(\mathbf{x}^{(r)})\|^2
\end{aligned}
\tag{57}
$$

where in $(a)$ we use the gradient Lipschitz continuity; $(b)$ is true because $\nabla f(\mathbf{x}^{(r)}) = \mathbf{P}(\mathbf{x}^{(r)})\nabla f(\mathbf{x}^{(r)}) + \mathbf{P}_\perp(\mathbf{x}^{(r)})\nabla f(\mathbf{x}^{(r)})$ and $q_\pi(\mathbf{x}^{(r)}) = \mathbf{P}(\mathbf{x}^{(r)})\nabla f(\mathbf{x}^{(r)})$. It can be observed that there must exist a small $\alpha$ such that the objective is decreased, so the line search algorithm will be terminated within finite number of steps.

Second, the definition of $\alpha_{\max}^{(r)}$ suggests that along the direction $-q_\pi(\mathbf{x}^{(r)})$, one can go as far as $\alpha_{\max}^{(r)}$ without being infeasible. Then we can determine a lower bound of $\alpha_{\max}^{(r)}$ before entering the line search step. To determine such a lower bound, we have two steps.

**Step (a)** Suppose that $\alpha_{\max}^{(r)}$ satisfies (26), that is $f(\mathbf{x}^{(r)} + \alpha_{\max}^{(r)}\mathbf{d}^{(r)}) > f(\mathbf{x}^{(r)}) + \frac{1}{2}\rho(\alpha_{\max}^{(r)})$. Then we must have $\alpha_{\max}^{(r)} \geq \frac{1}{L_1}$, because otherwise, we have

$$
\alpha_{\max}^{(r)} \leq \frac{1}{L_1} \Rightarrow (\alpha_{\max}^{(r)} - \frac{(\alpha_{\max}^{(r)})^2 L_1}{2}) \geq \frac{1}{2}\alpha_{\max}^{(r)}
$$

The above fact combined with the descent estimate (57) implies that (26) stops to hold true, which is a contradiction.

**Step (b)** Suppose that $\alpha_{\max}^{(r)}$ does not satisfy (26). This implies that line 8 of the line search algorithm will hold, so the algorithm has already returned. This is a contradiction to the assumption that the line search step will be performed. Therefore we conclude that the initial step-size $\alpha_{\max}^{(r)}$ before entering the line search procedure is lower bounded by $1/L_1$.

Third, we will show that the line search terminates with $\alpha \geq 1/(2L_1)$ and has the sufficient descent. By adopting the line search algorithm and applying stopping criteria (26), we know that $\alpha$ will terminate in the interval $[1/(2L_1), 1/L_1]$. To show the minimum descent of the objective value, note the following series of inequalities

$$
f(\mathbf{x}^{(r+1)}) \overset{(a)}{\leq} f(\mathbf{x}^{(r)}) - \frac{3}{8L_1}\|q_\pi(\mathbf{x}^{(r)})\|^2 \overset{(b)}{<} f(\mathbf{x}^{(r)}) - \frac{3}{8}\frac{\epsilon_H'^3(\delta)}{L_2^2},
\tag{58}
$$

where in $(a)$ we substitute $\alpha = 1/(2L_1)$ into (57) due to the monotonic behavior of the function $\alpha - \alpha^2 L_1/2$ over the interval $[1/(2L_1), 1/L_1]$; in $(b)$ we use $\|q_\pi(\mathbf{x}^{(r)})\|^2 \geq 135 L_1 \epsilon_H'^3(\delta)/(128 L_2^2)$ since in line 8 of Algorithm 1 we know from the algorithm that $-q_\pi(\mathbf{x}^{(r)})$ is chosen when

$$
-q_\pi(\mathbf{x}^{(r)})^T \mathbf{v}(\mathbf{x}^{(r)}) \frac{3L_1 \epsilon_H'(\delta)}{L_2} + \frac{135 L_1 \epsilon_H'^3(\delta)}{128 L_2^2} \leq \|q_\pi(\mathbf{x}^{(r)})\|^2, \quad q_\pi(\mathbf{x}^{(r)})^T \mathbf{v}(\mathbf{x}^{(r)}) \leq 0.
$$

This completes the proof. $\qquad\square$

**Lemma 6.** *Suppose $\mathbf{d}^{(r)}$ is chosen by $\mathbf{v}(\mathbf{x}^{(r)})$, $\mathbf{x}^{(r+1)}$ is computed by the NCD procedure in Algorithm 1 and line 7 of Algorithm 2 does not hold, i.e., $f(\mathbf{x}^{(r)} + \alpha_{\max}^{(r)}\mathbf{d}^{(r)}) \geq f(\mathbf{x}^{(r)})$. Then, $\alpha_{\max}^{(r)} \geq \frac{9\epsilon_H'(\delta)}{4L_2}$ and the line search terminates with $\alpha \geq 9\epsilon_H'(\delta)/(8L_2)$, and we have: $f(\mathbf{x}^{(r+1)}) \leq f(\mathbf{x}^{(r)}) - 3\epsilon_H'^3(\delta)/(8L_2^2)$.*

*Proof.* Suppose $\mathbf{d}^{(r)} = \mathbf{v}(\mathbf{x}^{(r)})$ in line 10 of Algorithm 1. Then according to the $L_2$-Lipschitz continuity assumption, we have

$$f(\mathbf{x}^{(r)} + \alpha \mathbf{v}(\mathbf{x}^{(r)}))$$
$$\leq f(\mathbf{x}^{(r)}) + \alpha q_\pi(\mathbf{x}^{(r)})^T \mathbf{v}(\mathbf{x}^{(r)}) + \frac{\alpha^2}{2}\mathbf{v}(\mathbf{x}^{(r)})^T \nabla^2 f(\mathbf{x}^{(r)})\mathbf{v}(\mathbf{x}^{(r)}) + \frac{\alpha^3}{6}L_2\|\mathbf{v}(\mathbf{x}^{(r)})\|^3$$
$$= f(\mathbf{x}^{(r)}) + \alpha q_\pi(\mathbf{x}^{(r)})^T \mathbf{v}(\mathbf{x}^{(r)}) + \frac{\alpha^2}{2}\mathbf{v}(\mathbf{x}^{(r)})^T \nabla^2 f(\mathbf{x}^{(r)})\mathbf{v}(\mathbf{x}^{(r)}) + \frac{\alpha^3}{6}L_2, \tag{59}$$

where we used $\|\mathbf{v}(\mathbf{x}^{(r)})\| = 1$. Since the following hold $q_\pi(\mathbf{x}^{(r)})^T \mathbf{v}(\mathbf{x}^{(r)}) \leq 0$
$\mathbf{v}(\mathbf{x}^{(r)})^T \nabla^2 f(\mathbf{x}^{(r)})\mathbf{v}(\mathbf{x}^{(r)}) \leq -\epsilon'_H(\delta)$, we know that

$$\alpha q_\pi(\mathbf{x}^{(r)})^T \mathbf{v}(\mathbf{x}^{(r)}) + \frac{\alpha^2}{2}\mathbf{v}(\mathbf{x}^{(r)})^T \nabla^2 f(\mathbf{x}^{(r)})\mathbf{v}(\mathbf{x}^{(r)}) \leq -\frac{\alpha^2 \epsilon'_H(\delta)}{2} < 0. \tag{60}$$

Then, combining (59) and (60) we obtain

$$f(\mathbf{x}^{(r)} + \alpha \mathbf{v}(\mathbf{x}^{(r)})) - f(\mathbf{x}^{(r)}) \leq -\left(1 - \frac{\alpha L_2}{3\epsilon'_H(\delta)}\right)\frac{\alpha^2 \epsilon'_H(\delta)}{2}. \tag{61}$$

It follows that when choosing $0 < \alpha < \frac{3\epsilon'_H(\delta)}{L_2}$, the objective function is decreasing. By using the similar argument as in the previous lemma (by applying criteria (26)), we can conclude that the initial $\alpha_{\max}^{(r)}$ before entering the line search procedure will satisfy:

$$\alpha_{\max}^{(r)} \geq \frac{9\epsilon'_H(\delta)}{4L_2}. \tag{62}$$

Finally, it is easy to see that by using the backtracking line search where each time the step-size is shrunk by $1/2$, the algorithm will stop at $\alpha \geq 9\epsilon'_H(\delta)/(8L_2)$, therefore we will have at least the following amount of descent:

$$f(\mathbf{x}^{(r+1)}) \leq f(\mathbf{x}^{(r)}) - \frac{5}{8}\frac{(\frac{9}{8})^2 \epsilon'^3_H(\delta)}{2L_2^2} < f(\mathbf{x}^{(r)}) - \frac{3}{8}\frac{\epsilon'^3_H(\delta)}{L_2^2}, \tag{63}$$

which completes the proof. $\qquad\square$

**Lemma 7.** *Algorithm 1 will stop if for $\min\{d, m\}$ consecutive iterations, its line search procedure only returns with step-size $\alpha_{\max}^{(r)}$ chosen as in (23).*

*Proof.* First, we show that $\dim(\mathbf{x}^{(r)})$ is not increasing if $\mathbf{x}^{(r)}$ is updated by NCD successively. Since at the $r$th iteration, the equality $\mathbf{A}_\mathcal{A}(\mathbf{x}^{(r)})\mathbf{x}^{(r)} = \mathbf{b}_\mathcal{A}(\mathbf{x}^{(r)})$ holds (due to the definition of active set), which implies that

$$\mathbf{A}_\mathcal{A}(\mathbf{x}^{(r)})\mathbf{x}^{(r+1)} = \mathbf{A}_\mathcal{A}(\mathbf{x}^{(r)})\left(\mathbf{x}^{(r)} + \alpha^{(r)}\mathbf{d}^{(r)}\right) = \mathbf{b}_\mathcal{A}(\mathbf{x}^{(r)}) + \alpha^{(r)}\mathbf{A}_\mathcal{A}(\mathbf{x}^{(r)})\mathbf{d}^{(r)} \overset{(a)}{=} \mathbf{b}_\mathcal{A}(\mathbf{x}^{(r)}) \tag{64}$$

where $(a)$ is true because $\mathbf{d}^{(r)} \in \mathsf{Null}(\mathbf{A}_\mathcal{A}(\mathbf{x}^{(r)}))$, so $\dim(\mathcal{F}(\mathbf{x}^{(r+1)}))$ is no more than $\dim(\mathcal{F}(\mathbf{x}^{(r)}))$. Second, we show that if $\alpha_{\max}^{(r)}$ is chosen, $\dim(\mathcal{F}(\mathbf{x}^{(r)}))$ is decreased at least by 1. Since at the $r + 1$th iteration the algorithm still chooses $\alpha_{\max}^{(r+1)}$, meaning that iterate $\mathbf{x}^{(r+1)}$ has at least one new active constraint, i.e., $\dim(\mathcal{F}(\mathbf{x}^{(r)})) \geq \dim(\mathcal{F}(\mathbf{x}^{(r+1)})) + 1$. In other words, when step-size $\alpha_{\max}^{(r)}$ is chosen and updated by (23), the dimension of the free space is reduced at least by 1. Therefore, if step-size $\alpha_{\max}^{(r)}$ is chosen consecutively and updated by (23), $\dim(\mathcal{F}(\mathbf{x}^{(r)}))$ is monotonically decreasing. Since the dimension of the subspace is at most $d$ and the total number of constraints is at most $m$, the algorithm consecutively performs NCD at most $\min\{d, m\}$ times. $\quad\square$

## C.2 Simplified SNAP

Before proving Theorem 1, we provide a simplified version of SNAP shown in Algorithm 3 and show the convergence of this algorithm, which will be helpful of understanding the key steps in the proof of SNAP. The reason is that some techniques, which are considered in SNAP to reduce the computational complexity, involve multiple branches that SNAP may use. A combinatorial choice of these subroutines makes the convergence analysis complicated, so it will be more intuitive to see the proof for the simplified algorithm, which essentially has the same rate as SNAP. Here, we give a concise proof for Algorithm 3 in the following.

---

**Algorithm 3** A *simplified* Successive Negative-curvature grAdient Projection (sSNAP) algorithm

---
1: **Input:** $\mathbf{x}^{(1)}, \epsilon_G, \epsilon_H, L_1, L_2, \alpha_\pi = 1/L_1, \delta, \mathbf{A}, \mathbf{b}, \mathsf{flag} = \lozenge$
2: **for** $r = 1, \ldots$ **do**
3:     **if** $\|g_\pi(\mathbf{x}^{(r)})\| \leq \epsilon_G$ **then**
4:         $[\mathsf{flag}, \mathbf{v}(\mathbf{x}^{(r)}), -\epsilon'_H(\delta)] = $ *Negative-Eigen-Pair*$(\mathbf{x}^{(r)}, f, \delta)$
5:         **if** $\mathsf{flag} = \lozenge$ **then**
6:             Compute $q_\pi(\mathbf{x}^{(r)})$ by (6)
7:             Choose $\mathbf{v}(\mathbf{x}^{(r)})$ such that $q_\pi(\mathbf{x}^{(r)})^T \mathbf{v}(\mathbf{x}^{(r)}) \leq 0$
8:             $\mathbf{d}^{(r)} = \mathbf{v}(\mathbf{x}^{(r)})$                  ▷ Choose negative curvature direction
9:             Update $\mathbf{x}^{(r+1)}$ by Algorithm 2                 ▷ Perform line search
10:         **else**
11:             Output $\mathbf{x}^{(r)}$
12:         **end if**
13:     **else**
14:         Update $\mathbf{x}^{(r+1)}$ by (13)                       ▷ Perform PGD
15:     **end if**
16: **end for**

---

*Proof.* We will show that after a number of iterations given by Algorithm 3, the algorithm will converge to an $(\epsilon_G, \epsilon_H)$-SOSP1 defined in (8).

Let us suppose that at a given point $\mathbf{x}^{(r)}$, the condition (8) does not hold.

First suppose that the first-order condition is not satisfied, that is $\|g_\pi(\mathbf{x}^{(r)})\| \geq \epsilon_G$. Then the algorithm will perform the PGD step (13). By Lemma 4, the descent of the objective value is given by $\frac{\epsilon_G^2}{18L_1}$.

Second, when the size of the gradient is small, but the second-order condition in (8b) is not satisfied (i.e., when $\mathsf{flag} = \lozenge$). Then in this case, NCD will be performed, and there are two choices for selecting the step-size:

**Case 1) ($\mathsf{flag}_\alpha = \emptyset$):** The algorithm implements $\mathbf{x}^{(r+1)} = \mathbf{x}^{(r)} + \alpha^{(r)}\mathbf{d}^{(r)}$ without using $\alpha_{\max}^{(r)}$ computed by (23).

**Case 2) ($\mathsf{flag}_\alpha = \lozenge$):** $\alpha_{\max}^{(r)}$ is computed by (23) to update $\mathbf{x}^{(r+1)}$.

In the first case, we know that if $\alpha_{\max}^{(r)}$ is not chosen by (23), then some sufficient descent will be achieved. From Lemma 5, we know that after one step update the objective value decreases as

$$f(\mathbf{x}^{(r+1)}) \leq f(\mathbf{x}^{(r)}) - \Delta, \text{ where } \Delta = \frac{3\epsilon_H'^3(\delta)}{8L_2^2}. \tag{65}$$

In the second case, the descent for each step may not be quantified. However, it is important to see that, by Lemma 7, the algorithm can repeat this case (i.e., choosing $\alpha_{\max}^{(r)}$ by (23)) for at most $\min\{d, m\}$ consecutive times.

By using the above fact, let us look at the second case in more detail and see how we can quantify the descent achieved by some $k \leq \min\{d, m\}$ consecutive times that **Case 2)** happens. Since **Case 2)** can happen at most $\min\{d, m\}$ consecutively times, our strategy is to trace back the steps of the algorithm from the current iteration $\mathbf{x}^{(r)}$ and see what happens. To this end, let us suppose that at iteration $r$ **Case 2)** happens.

First of all, if the sequence has never been updated by either **Case 1** or PGD, the algorithm must stop by at most $d$ iterations. If the algorithm stops, it is clear that an $(\epsilon_G, \epsilon_H)$-SOSP1 solution is obtained. This is because the inactive set becomes empty and (8b) is satisfied automatically.

Second, consider iteration from $r - \min\{d, m\}$ until $r$. The sequence must be updated by either **Case 1)** or the PGD step, otherwise the algorithm will stop and output an $(\epsilon_G, \epsilon_H)$-SOSP1 solution. Then we must have

$$f(\mathbf{x}^{(r)}) - f(\mathbf{x}^{(r-\min\{d,m\})}) < -\min\left\{\epsilon_G^2/(18L_1), 3\epsilon_H'^3(\delta)/(8L_2^2)\right\}, \forall r > \min\{d, m\}. \quad (66)$$

Summarizing the argument so far, we have that, after every consecutive $\min\{d, m\}$ iterations of the algorithm, either the algorithms stops, or (66) holds true.

After applying the telescoping sum on (66), we have

$$f^\star - f(\mathbf{x}^{(1)}) \le f(\mathbf{x}^{(r)}) - f(\mathbf{x}^{(1)}) \le -r\frac{\min\left\{\epsilon_G^2/(18L_1), 3\epsilon_H'^3(\delta)/(8L_2^2)\right\}}{\min\{d, m\}}. \quad (67)$$

where $f^\star$ denotes the minimum objective value achieved by the global optimal solution. By defining

$$\Delta' \triangleq \min\left\{\frac{3\epsilon_H'^3(\delta)}{8L_2^2}, \frac{\epsilon_G^2}{18L_1}\right\} \frac{1}{\min\{d, m\}}, \quad (68)$$

we obtain

$$r \le \frac{f(\mathbf{x}^{(1)}) - f^\star}{\Delta'}. \quad (69)$$

Since the probability that eigen-pair fails to extract the negative curvature is $\delta$, applying the union bound, we only need to set $\delta' = \delta(f(\mathbf{x}^{(1)} - f^\star)/\Delta'$ so that we can have the claim that SNAP will output approximate SOSP1s with probability $1 - \delta'$. Note that $\gamma\epsilon_H'(\delta) > \epsilon_H$. We can obtain the convergence rate of Algorithm 3 by

$$\widetilde{\mathcal{O}}\left(\frac{\min\{d, m\}(f(\mathbf{x}^{(1)}) - f^\star)}{\min\left\{\frac{\epsilon_G^2}{L_1}, \frac{\epsilon_H^3}{L_2^2}\right\}}\right), \quad (70)$$

which completes the proof. □

## C.3  Proof of Theorem 1

Compared with the simplified SNAP, SNAP has two main differences: 1) $\mathbf{d}^{(r)}$ can be chosen by either $-q_\pi(\mathbf{x}^{(r)})$ or $\mathbf{v}(\mathbf{x}^{(r)})$ in the NCD step based on the minimum amount of the objective reduction; 2) when $\mathsf{flag}_\alpha^{(r)} = \emptyset$ there is a minimum number of iterations (denoted by $r_\mathsf{th}$) that SNAP calls subroutine *Negative-Eigen-Pair* twice.

*Proof.* First, suppose that at a given point $\mathbf{x}^{(r)}$, the condition (8) does not hold. If the first-order condition is not satisfied, (i.e., $\|g_\pi(\mathbf{x}^{(r)})\| \ge \epsilon_G$), then by Lemma 4, we can show that performing the PGD step (13) achieves the following descent:

$$f(\mathbf{x}^{(r+1)}) \le f(\mathbf{x}^{(r)}) - \frac{\epsilon_G^2}{18L_1}. \quad (71)$$

Second, when $\|g_\pi(\mathbf{x}^{(r)})\|$ is small, but the second-order condition (8b) is not satisfied. Then the NCD will be performed, and there are two choices of step-sizes:
**Case 1) ($\mathsf{flag}_\alpha = \emptyset$):** The algorithm implements $\mathbf{x}^{(r+1)} = \mathbf{x}^{(r)} + \alpha^{(r)}\mathbf{d}^{(r)}$ without using $\alpha_\mathrm{max}^{(r)}$ computed by (23).
**Case 2) ($\mathsf{flag}_\alpha = \Diamond$):** $\alpha_\mathrm{max}^{(r)}$ is computed by (23) to update $\mathbf{x}^{(r+1)}$.

In the **first** case, we know that if $\alpha_\mathrm{max}^{(r)}$ is not chosen by (23), then some sufficient descent will be achieved. In particular, from Lemma 5–Lemma 6, no matter which direction (i.e., either $-q_\pi(\mathbf{x}^{(r)})$ or $\mathbf{v}(\mathbf{x}^{(r)})$) is chosen, we have:

$$f(\mathbf{x}^{(r+1)}) \le f(\mathbf{x}^{(r)}) - \Delta, \text{ where } \Delta \triangleq \frac{3\epsilon_H'^3(\delta)}{8L_2^2}. \quad (72)$$

After performing one step, $\mathsf{flag}_\alpha$ becomes $\lozenge$. From the algorithm we know that $r_{th}$ number of PGD will be performed. However, the amount of descent cannot be quantified (because we are in NCD so $\|g_\pi(\mathbf{x}^{(r)})\| \le \epsilon_G$). Thus, we still have

$$f(\mathbf{x}^{(r+r_{th})}) \le f(\mathbf{x}^{(r)}) - \Delta. \tag{73}$$

In the **second** case, the descent for each step may not be quantified. However, a key result shown in Lemma 7 is that, the algorithm can repeat this case (i.e., choosing $\alpha_{\max}^{(r)}$ by (23)) for at most $\min\{d, m\}$ *consecutive* times. Based on such a key result, let us analyze the descent when **Case 2)** happens in $k \le \min\{d, m\}$ consecutive times. To this end, let us suppose that at iteration $r$ **Case 2)** happens.

First of all, if $d$ iterations of **Case 2)** are executed, then the algorithm must stop and $(\epsilon_G, \epsilon_H)$-SOSP1 solution is obtained. This is because the inactive set becomes empty and (8b) is satisfied automatically.

Second, consider the iterations in $[(r - \min\{d, m\}), \ r]$. The sequence must be updated by either **Case 1)** or the PGD step, otherwise the algorithm will stop and output an $(\epsilon_G, \epsilon_H)$-SOSP1 solution (as in the previous case). If either **Case 1)** or the PGD happens, we must have

$$f(\mathbf{x}^{(r)}) - f(\mathbf{x}^{(r-\min\{d,m\})}) \le -\min\left\{\epsilon_G^2/(18L_1), 3\epsilon_H'^3(\delta)/(8L_2^2)\right\}, \forall r > \min\{d, m\}. \tag{74}$$

Summarizing the argument so far, after every consecutive $\min\{d, m\}$ iterations of the algorithm, either the algorithm stops or (74) holds true.

Note that **Case 1** and **Case 2** are mutually exclusive. Take $T' \triangleq \min\{d, m\} \cdot r_{\mathsf{th}}$. From (72), we know that
$$f(\mathbf{x}^{(r+T')}) - f(\mathbf{x}^{(r)}) \le -\min\{d, m\}\frac{3\epsilon_H'^3(\delta)}{8L_2^2}. \tag{75}$$
From (74), we have

$$f(\mathbf{x}^{(r)}) - f(\mathbf{x}^{(r-T')}) \le -r_{\mathsf{th}}\min\left\{\frac{\epsilon_G^2}{18L_1}, \frac{3\epsilon_H'^3(\delta)}{8L_2^2}\right\}. \tag{76}$$

Putting (75) and (76) together, we have

$$f(\mathbf{x}^{(r+T')}) - f(\mathbf{x}^{(r-T')}) \le -\min\left\{\min\{d, m\}\frac{3\epsilon_H'^3(\delta)}{8L_2^2}, r_{\mathsf{th}}\min\left\{\frac{\epsilon_G^2}{18L_1}, \frac{3\epsilon_H'^3(\delta)}{8L_2^2}\right\}\right\}. \tag{77}$$

Let $n$ be the number of $2T'$ blocks contained in $[1, r]$. After applying the telescoping sum on (71), (77), we have

$$\begin{aligned}
f^\star - f(\mathbf{x}^{(1)}) \le & f(\mathbf{x}^{(r)}) - f(\mathbf{x}^{(1)}) \le f(\mathbf{x}^{(2nT'+1)}) - f(\mathbf{x}^{(1)}) \\
\le & -n\min\left\{\min\{d, m\}\frac{3\epsilon_H'^3(\delta)}{8L_2^2}, r_{\mathsf{th}}\min\left\{\frac{\epsilon_G^2}{18L_1}, \frac{3\epsilon_H'^3(\delta)}{8L_2^2}\right\}, T'\frac{\epsilon_G^2}{18L_1}\right\}
\end{aligned} \tag{78}$$

where $f^\star$ denotes the minimum objective value achieved by the global optimal solution, and $n \ge (r-1)/(2T')$. By defining

$$\Delta' \triangleq \min\left\{\min\{d, m\}\frac{3\epsilon_H'^3(\delta)}{8L_2^2}, r_{\mathsf{th}}\min\left\{\frac{\epsilon_G^2}{18L_1}, \frac{3\epsilon_H'^3(\delta)}{8L_2^2}\right\}, T'\frac{\epsilon_G^2}{18L_1}\right\}, \tag{79}$$

we obtain

$$n \le \frac{f(\mathbf{x}^{(1)}) - f^\star}{\Delta'}, \quad r \le 2nT' + 1 \le \frac{2T'(f(\mathbf{x}^{(1)}) - f^\star)}{\Delta'} + 1. \tag{80}$$

Since the probability that eigen-pair fails to extract the negative curvature is $\delta$, applying the union bound, we only need to set $\delta' = \delta(2T'(f(\mathbf{x}^{(1)}) - f^\star)/\Delta' + 1)$ so that we can have the claim that SNAP will output approximate SOSP1s with probability $1 - \delta'$.

Note that $\gamma\epsilon_H'(\delta) > \epsilon_H$. We can obtain the convergence rate of Algorithm 1 by (14), which completes the proof. $\qquad\square$

---

**Algorithm 4** Negative Curvature Extraction by Subspace Gradient Descent from Noise (SGDN)

---
1: **Input:** $\mathbf{x}^{(r)}, \mathcal{F}(\mathbf{x}^{(r)}), T, q_\pi, \mathscr{F}, \mathscr{R}, \beta = 1/L_1, d, \delta, \widehat{c}, \epsilon_H$
2: Generate $\mathbf{z}$ randomly from the sphere of an Euclidean ball of radius $\mathscr{R}$ in $\mathcal{F}(\mathbf{x}^{(r)})$.
3: **for** $\tau = 1, \dots, T$ **do**
$$\mathbf{z}^{(\tau+1)} = \mathbf{z}^{(\tau)} - \beta(q_\pi(\mathbf{x}^{(r)} + \mathbf{z}^{(\tau)}) - q_\pi(\mathbf{x}^{(r)})) \tag{81}$$
4: **end for**
5: **if** $f(\mathbf{x}^{(r)} + \mathbf{z}^{(T)}) - f(\mathbf{x}^{(r)}) - q_\pi(\mathbf{x}^{(r)})^T \mathbf{z}^{(T)} \leq -1.5\mathscr{F}$ **then**
6:     return $[\Diamond, \mathbf{z}^{(T)}/\|\mathbf{z}^{(T)}\|, -\frac{\epsilon_H}{8\widehat{c}\log(\frac{dL_1}{\epsilon_H \delta})}]$
7: **else**     return $[\emptyset, 0, 0]$
8: **end if**

---

# D    Proofs of SNAP$^+$

## D.1    Negative Curvature Extraction by Subspace Gradient Descent from Noise

The details of SGDN is shown in Algorithm 4.

## D.2    Technical Lemmas

Before showing the convergence analysis, let us first have the following definitions and show some basic properties of the SGDN iterates for the case where Assumption 1 holds.

**Condition 1.** *A strict saddle point* $\mathbf{x}$ *satisfies* $\lambda_{\min}(\mathbf{H_P}(\mathbf{x})) \leq -\epsilon_H$.

Note here it is not necessary to further require $\|g_\pi(\mathbf{x})\| \leq \epsilon_G$. Let $\vec{e}$ denote the eigenvector that corresponds to the smallest eigenvalue of $\mathbf{H_P}(\mathbf{x})$.

We define an *auxiliary function* as $\widehat{f}_{\mathbf{x}}(\mathbf{z}) \triangleq f(\mathbf{x} + \mathbf{z}) - f(\mathbf{x}) - \nabla_{\mathbf{x}} f(\mathbf{x})^T \mathbf{z}$ near a strict saddle point $\mathbf{x}$. Then, we can have $\nabla_{\mathbf{z}} \widehat{f}_{\mathbf{x}}(\mathbf{z}) = \nabla_{\mathbf{x}} f(\mathbf{x} + \mathbf{z}) - \nabla_{\mathbf{x}} f(\mathbf{x})$. It is easy to see that $\nabla_{\mathbf{z}} \widehat{f}_{\mathbf{x}}(\mathbf{z})$ is also $L_1$-Lipschitz continuous.

In the rest of the paper, we just use $\widehat{f}$ to denote $\widehat{f}_{\mathbf{x}}$ for simplicity. Let $\widehat{q}_\pi(\mathbf{z}) \triangleq \mathbf{P}\nabla_{\mathbf{z}}\widehat{f}(\mathbf{z})$ and $\mathbf{P}$ is the projection matrix defined in (5). Then we can have $\widehat{q}_\pi(\mathbf{z}) = q_\pi(\mathbf{x} + \mathbf{z}) - q_\pi(\mathbf{x})$.

From the update rule of SGDN (15), the iterates can be rewritten as

$$\mathbf{z}^{(r+1)} = \mathbf{z}^{(r)} - \beta\widehat{q}_\pi(\mathbf{z}^{(r)}). \tag{82}$$

Towards this end, we will give the following a series of lemmas that pave the way of showing the main convergence rate result. To be specific, Lemma 8 is a preliminary lemma which will be used in Lemma 9 and Theorem 2. Further combining Lemma 9 and Lemma 10 leads to Theorem 2.

**Lemma 8.** *If function* $f(\cdot)$ *is* $L_2$-*Hessian Lipschitz, we have*

$$\left\| \int_0^1 \mathbf{P}^T \nabla^2 f(\theta\mathbf{x})\mathbf{P} d\theta - \mathbf{P}^T \nabla^2 f(\mathbf{x}')\mathbf{P} \right\| \leq L_2\|\mathbf{x}\| + \|\mathbf{x}'\|, \forall \mathbf{x}, \mathbf{x}' \in \mathcal{X}. \tag{83}$$

*where* $\mathbf{P}$ *denotes the projection matrix and* $\theta \in [0, 1]$.

*Proof.* We have the following relation:

$$\left\| \int_0^1 \mathbf{P}^T \left( \nabla^2 f(\theta\mathbf{x}) - \nabla^2 f(\mathbf{x}') \right) \mathbf{P} d\theta \right\|$$

$$\overset{(a)}{\leq} \int_0^1 \|\nabla^2 f(\theta\mathbf{x}) - \nabla^2 f(\mathbf{x}')\| d\theta$$

$$\overset{(b)}{\leq} L_2 \int_0^1 \|\theta\mathbf{x} - \mathbf{x}'\| d\theta \leq L_2 \int_0^1 \theta\|\mathbf{x}\| d\theta + L_2\|\mathbf{x}'\| \leq L_2(\|\mathbf{x}\| + \|\mathbf{x}'\|)$$

where in $(a)$ we use Cauchy-Schwarz inequality and $\|\mathbf{P}\| = 1$; in $(b)$ we use the $L_2$-Hessian Lipschitz continuity. $\qquad\square$

To quantify the "sufficient" descent of SGDN when the iterates are around the strict saddle points, we also need to introduce some constants defined as follows,

$$\mathscr{F} \triangleq \frac{\epsilon_H^3}{L_2^2 \widehat{c}^5} \log^{-3}\left(\frac{d\kappa}{\delta}\right), \tag{84a}$$

$$\mathscr{S} \triangleq \frac{\epsilon_H}{L_2 \widehat{c}^2} \log^{-1}\left(\frac{d\kappa}{\delta}\right), \tag{84b}$$

$$\mathscr{T} \triangleq \frac{\log\left(\frac{d\kappa}{\delta}\right)}{\beta \epsilon_H}. \tag{84c}$$

These quantities refer to different units of the algorithm. Specifically, $\mathscr{F}$ accounts for the objective value, $\mathscr{S}$ for the norm of the difference between iterates, and $\mathscr{T}$ for the number of iterations. Also, we define a condition number in terms of $\epsilon_H$ as:

$$\kappa \triangleq \frac{L_1}{\epsilon_H} \geq 1. \tag{85}$$

In the process of the proofs, we also use conditions $\log(\frac{d\kappa}{\delta}) \geq 1$ when $\delta \in (0, \frac{d\kappa}{e}]$ repeatedly to simplify the expressions of the inequalities, where $e$ stands for the natural logarithm constant.

The following Lemma 9 and Lemma 10 characterize the descent of SGDN around the strict saddle points.

**Lemma 9.** *Under Assumption 1, consider* $\mathbf{x}$ *that satisfies Condition 1 and a sequence* $\mathbf{u}^{(r)}$ *generated by SGDN. Define* $\beta \leq 1/L_1$, *and the following:*

$$\mathscr{R} \triangleq \frac{\mathscr{S}}{\widehat{c}^2 \kappa \log\left(\frac{d\kappa}{e}\right)}, \quad and \quad T \triangleq \min\left\{\min_{r \geq 1}\{r | \widehat{f}(\mathbf{u}^{(r)}) - \widehat{f}(\mathbf{u}^{(1)}) \leq -2\mathscr{F}\}, \widehat{c} \cdot \mathscr{T}\right\}. \tag{86}$$

*Then for any constant* $\widehat{c} \geq \sqrt{2}$, $\delta \in (0, \frac{d\kappa}{e}]$, *when initial point* $\mathbf{u}^{(1)}$ *satisfies*

$$\|\mathbf{u}^{(1)} - \mathbf{x}\| \leq 2\mathscr{R}, \tag{87}$$

*the iterates generated by SGDN satisfy* $\|\mathbf{u}^{(r)} - \mathbf{u}^{(1)}\| \leq 2\mathscr{S}$ *and* $\|\mathbf{u}^{(r)} - \mathbf{x}\| \leq 3\mathscr{S}, \forall r < T$.

*Proof.* Without loss of generality, let $\mathbf{u}^{(1)}$ be the origin, i.e., $\mathbf{u}^{(1)} = 0$. According to the update rule of SGDN, we have

$$\mathbf{u}^{(r+1)} = \mathbf{u}^{(r)} - \beta \widehat{q}_\pi(\mathbf{u}^{(r)}). \tag{88}$$

Similar as the derivation in (57), by the $L_1$-gradient Lipschitz continuity, we have

$$\widehat{f}(\mathbf{u}^{(r+1)}) = \widehat{f}\left(\mathbf{u}^{(r)} - \beta \widehat{q}_\pi(\mathbf{x}^{(r)})\right) \leq \widehat{f}(\mathbf{u}^{(r)}) - \beta \nabla \widehat{f}(\mathbf{u}^{(r)})^T \widehat{q}_\pi(\mathbf{u}^{(r)}) + \frac{\beta^2}{2} L_1 \|\widehat{q}_\pi(\mathbf{u}^{(r)})\|^2$$

$$\leq \widehat{f}(\mathbf{u}^{(r)}) - \beta \widehat{q}_\pi(\mathbf{u}^{(r)})^T \widehat{q}_\pi(\mathbf{u}^{(r)}) + \frac{\beta^2}{2} L_1 \|\widehat{q}_\pi(\mathbf{u}^{(r)})\|^2$$

$$= \widehat{f}(\mathbf{u}^{(r)}) - \left(\beta - \frac{\beta^2}{2} L_1\right) \|\widehat{q}_\pi(\mathbf{u}^{(r)})\|^2. \tag{89}$$

From (89), we also know that

$$\widehat{f}(\mathbf{u}^{(r+1)}) \overset{(a)}{\leq} \widehat{f}(\mathbf{u}^{(r)}) - \frac{\beta}{2} \|\widehat{q}_\pi(\mathbf{u}^{(r)})\|^2$$

$$\overset{(88)}{=} \widehat{f}(\mathbf{u}^{(r)}) - \frac{1}{2\beta} \|\mathbf{u}^{(r+1)} - \mathbf{u}^{(r)}\|^2 \tag{90}$$

where in $(a)$ we choose $\beta \leq 1/L_1$.

By applying the telescoping sum of (90), we have

$$\widehat{f}(\mathbf{u}^{(r+1)}) \leq \widehat{f}(\mathbf{u}^{(1)}) - \frac{1}{2\beta} \sum_{\tau=1}^{r} \|\mathbf{u}^{(\tau+1)} - \mathbf{u}^{(\tau)}\|^2, \quad \forall r < T. \tag{91}$$

According to the definition of $T$, we know that

$$\widehat{f}(\mathbf{u}^{(1)}) - \widehat{f}(\mathbf{u}^{(r)}) < 2\mathscr{F}, \quad \forall r < T. \tag{92}$$

Combining (91) and (92) , we know that

$$\sum_{\tau=1}^{r-1} \|\mathbf{u}^{(\tau+1)} - \mathbf{u}^{(\tau)}\|^2 < 4\beta\mathscr{F}. \tag{93}$$

Next, we will get the upper bound of $\|\mathbf{u}^{(r)} - \mathbf{u}^{(1)}\|, \forall r < T$ as the following. First, by the triangle inequality, we have

$$\|\mathbf{u}^{(r)} - \mathbf{u}^{(1)}\|^2 \leq (r-1)\sum_{\tau=1}^{r-1} \|\mathbf{u}^{(\tau+1)} - \mathbf{u}^{(\tau)}\|^2 \leq (T-1)\sum_{\tau=1}^{r-1} \|\mathbf{u}^{(\tau+1)} - \mathbf{u}^{(\tau)}\|^2 \tag{94}$$

$$\overset{(93)}{\leq} T4\beta\mathscr{F} \overset{(86)}{\leq} 4\widehat{c}\beta\mathscr{F}\mathscr{T} \overset{(a)}{\leq} 4\mathscr{S}^2, \tag{95}$$

where in $(a)$ we use the relation $\widehat{c}\beta\mathscr{F}\mathscr{T} = \mathscr{S}^2$ by applying (84a)(84b)(84c).

Due to the following fact

$$\|\mathbf{u}^{(r)} - \mathbf{x}\| = \|\mathbf{u}^{(r)} - \mathbf{u}^{(1)} + \mathbf{u}^{(1)} - \mathbf{x}\| \leq \underbrace{\|\mathbf{u}^{(r)} - \mathbf{u}^{(1)}\|}_{\leq 2\mathscr{S}} + \underbrace{\|\mathbf{u}^{(1)} - \mathbf{x}\|}_{\leq \frac{\mathscr{S}}{\widehat{c}^2 \log(\frac{d\kappa}{\delta})}} \leq 3\mathscr{S} \tag{96}$$

where the last inequality is true when $\widehat{c} \geq 1$, and $d\kappa/\delta > e$. Therefore, we know that $\|\mathbf{u}^{(r)} - \mathbf{x}\| \leq 3\mathscr{S}, \forall r < T$ where $\beta \leq 1/L_1$, which completes the proof. $\square$

**Lemma 10.** *Consider* $\mathbf{x}$ *that satisfies Condition 1. Suppose that there exist two sequences* $\{\mathbf{u}^{(r)}\}$ *and* $\{\mathbf{w}^{(r)}\}$*, generated by SGDN with two different initial points* $\{\mathbf{u}^{(1)}, \mathbf{w}^{(1)}\}$ *that satisfy*

$$\|\mathbf{u}^{(1)} - \mathbf{x}\| \leq \mathscr{R}, \ \mathbf{w}^{(1)} = \mathbf{u}^{(1)} + \upsilon\mathscr{R}\vec{\mathbf{e}}, \ \upsilon \in [\delta/(2\sqrt{d}), 1], \tag{97}$$

*where* $\mathscr{R}$ *is defined in* (86)*. Let us also define*

$$T \triangleq \min\left\{\min_{r \geq 1}\{r | \widehat{f}(\mathbf{w}^{(r)}) - \widehat{f}(\mathbf{w}^{(1)}) \leq -2\mathscr{F}\}, \widehat{c} \cdot \mathscr{T}\right\}. \tag{98}$$

*Suppose* $\widehat{c} \geq 51$*,* $\delta \in (0, \frac{d\kappa}{e}]$*,* $\beta \leq 1/L_1$*,* $\|\mathbf{u}^{(r)} - \mathbf{x}\| \leq 3\mathscr{S}, \forall r < T$*, then we will have* $T < \widehat{c} \cdot \mathscr{T}$*, that is, we must have*

$$\widehat{f}(\mathbf{w}^{(r)}) - \widehat{f}(\mathbf{w}^{(1)}) \leq -2\mathscr{F}. \tag{99}$$

*Proof.* Let $\mathcal{H} \triangleq \mathbf{H}_{\mathbf{P}}(\mathbf{x})$, where $\mathbf{x}$ satisfies Condition 1. Without loss of generality, let $\mathbf{u}^{(1)} = 0$ and define $\mathbf{v}^{(r)} \triangleq \mathbf{w}^{(r)} - \mathbf{u}^{(r)}$. According to the assumption of Lemma 10, we know

$$\mathbf{v}^{(1)} = \mathbf{w}^{(1)} = \upsilon\mathscr{R}\vec{\mathbf{e}} = \upsilon\frac{\mathscr{S}}{\widehat{c}^2\kappa \log\left(\frac{d\kappa}{e}\right)}\vec{\mathbf{e}} \tag{100}$$

where $\upsilon \in [\delta/(2\sqrt{d}), 1]$. Clearly we have $\|\mathbf{v}^{(1)}\| \leq \mathscr{R}$. From (82), it is clear that $\mathbf{u}^{(r)} \in \mathcal{F}(\mathbf{x}^{(r)})$, i.e., $\mathbf{P}\mathbf{u}^{(r)} = \mathbf{u}^{(r)}$. Since $\mathbf{w}^{(r)}$ is also generated by SGDN and the sequence is initialized in $\mathcal{F}(\mathbf{x})$ as shown in (97), it is obvious that $\mathbf{w}^{(r)} \in \mathcal{F}(\mathbf{x})$, i.e., $\mathbf{P}\mathbf{w}^{(r)} = \mathbf{w}^{(r)}$. Thus, we have $\mathbf{v}^{(r)} \in \mathcal{F}(\mathbf{x})$,

i.e., $\mathbf{P}\mathbf{v}^{(r)} = \mathbf{v}^{(r)}$. Then sequence $\mathbf{w}^{(r+1)}$ can be expressed by

$$
\begin{aligned}
&\mathbf{u}^{(r+1)} + \mathbf{v}^{(r+1)} \\
=&\mathbf{w}^{(r+1)} & (101) \\
=&\mathbf{w}^{(r)} - \beta\left(q_\pi(\mathbf{w}^{(r)} + \mathbf{x}) - q_\pi(\mathbf{x})\right) \\
=&\mathbf{u}^{(r)} + \mathbf{v}^{(r)} - \beta\left(q_\pi(\mathbf{u}^{(r)} + \mathbf{v}^{(r)} + \mathbf{x}) - q_\pi(\mathbf{x})\right) \\
=&\mathbf{u}^{(r)} + \mathbf{v}^{(r)} - \beta\left(q_\pi(\mathbf{u}^{(r)} + \mathbf{v}^{(r)} + \mathbf{x}) - q_\pi(\mathbf{u}^{(r)} + \mathbf{x}) + q_\pi(\mathbf{u}^{(r)} + \mathbf{x}) - q_\pi(\mathbf{x})\right) \\
\overset{(a)}{=}&\mathbf{u}^{(r)} - \beta\left(q_\pi(\mathbf{u}^{(r)} + \mathbf{x}) - q_\pi(\mathbf{x})\right) + \mathbf{v}^{(r)} - \beta\left[\int_0^1 \mathbf{P}^T \nabla^2 f(\mathbf{u}^{(r)} + \mathbf{x} + \theta\mathbf{v}^{(r)})\mathbf{P}d\theta\right]\mathbf{v}^{(r)} \\
=&\mathbf{u}^{(r)} - \beta\left(q_\pi(\mathbf{u}^{(r)} + \mathbf{x}) - q_\pi(\mathbf{x})\right) + \mathbf{v}^{(r)} - \beta(\mathcal{H} + \Delta^{(r)})\mathbf{v}^{(r)} \\
=&\mathbf{u}^{(r+1)} + (\mathbf{I} - \beta\mathcal{H} - \beta\Delta^{(r)})\mathbf{v}^{(r)} & (102)
\end{aligned}
$$

where $(a)$ uses the Mean Value Theorem and $\mathbf{P}\mathbf{v}^{(r)} = \mathbf{v}^{(r)}$; $\Delta^{(r)} = \int_0^1 \mathbf{P}^T \nabla^2 f(\mathbf{u}^{(r)} + \mathbf{x} + \theta\mathbf{v}^{(r)})d\theta - \mathcal{H}$. Therefore, we have

$$
\mathbf{v}^{(r+1)} = (\mathbf{I} - \beta\mathcal{H} - \beta\Delta^{(r)})\mathbf{v}^{(r)}. \tag{103}
$$

By applying Lemma 8 and $L_2$-Lipschitz continuity of $f(\cdot)$, we have

$$
\|\Delta^{(r)}\| \le L_2(\|\mathbf{u}^{(r)}\| + \|\mathbf{v}^{(r)}\| + 2\|\mathbf{x}\|). \tag{104}
$$

Note that $\|\mathbf{w}^{(1)} - \mathbf{x}\| \le \|\mathbf{u}^{(1)} - \mathbf{x}\| + \|\mathbf{v}^{(1)}\| \le 2\mathcal{R}$. This means that as a sequence generated by SGDN, $\{\mathbf{w}^{(r)}\}$ satisfies the assumption given in Lemma 9. Also note that we have assumed that $\widehat{c} \ge 51$, then by the same lemma, it follows that

$$
\|\mathbf{w}^{(r)} - \mathbf{x}\| \le 3\mathcal{S}, \quad \forall r < T.
$$

Similarly, we can apply Lemma 9 again to obtain $\|\mathbf{u}^{(r)} - \mathbf{x}\| \le 3\mathcal{S}, \forall r < T$ since we have assumed $\|\mathbf{u}^{(1)} - \mathbf{x}\| \le \mathcal{R}$. Combining these two results, we have

$$
\|\mathbf{v}^{(r)}\| = \|\mathbf{w}^{(r)} - \mathbf{u}^{(r)}\| \le \|\mathbf{w}^{(r)} - \mathbf{x}\| + \|\mathbf{u}^{(r)} - \mathbf{x}\| \le 6\mathcal{S}. \tag{105}
$$

Next let us prove that the following hold: $\|\mathbf{x}\| \le \mathcal{R} \le \mathcal{S}$, where the first inequality is because the assumption that $\mathbf{u}^{(1)} = 0$ and $\|\mathbf{u}^{(1)} - \mathbf{x}\| \le \mathcal{R}$; the second inequality is due to the following choices of the constants $\widehat{c} \ge 1$, $\kappa \ge 1$ and $\log(d\kappa/\delta) \ge 1$. Further, from (95) and the assumption that $\mathbf{u}^{(1)} = 0$, we have $\|\mathbf{u}^{(r)}\| \le 2\mathcal{S}$. Combining the above relations with (104), we conclude

$$
\|\Delta^{(r)}\| \le 10L_2\mathcal{S} \quad \text{and} \quad \beta\|\Delta^{(r)}\| \le 10\beta L_2\mathcal{S}. \tag{106}
$$

By Condition 1 we know that $\mathbf{I} - \beta\mathcal{H}$ has maximum eigenvalue at least $1 + \epsilon_H\beta$. Let $\phi^{(r)}$ denote the norm of $\mathbf{v}^{(r)}$ projected on the space spanned by $\vec{\mathbf{e}}$, and let $\psi^{(r)}$ denote the norm of $\mathbf{v}^{(r)}$ projected onto the remaining space. From (103), we have

$$
\phi^{(r+1)} \ge (1 + \epsilon_H\beta)\phi^{(r)} - \mu\sqrt{(\phi^{(r)})^2 + (\psi^{(r)})^2}, \tag{107a}
$$

$$
\psi^{(r+1)} \le (1 + \epsilon_H\beta)\psi^{(r)} + \mu\sqrt{(\phi^{(r)})^2 + (\psi^{(r)})^2}, \tag{107b}
$$

where we have defined

$$
\mu = 10\beta L_2\mathcal{S}, \tag{108}
$$

and the inequalities are true due to the use of triangular inequality and the bound in (106). Then, we will use mathematical induction to prove

$$
\psi^{(r)} \le 4\mu r\phi^{(r)}, \ \forall\, r < T. \tag{109}
$$

Intuitively, the above result says that, the projection of $\mathbf{v}^{(r)}$ in the negative curvature direction should be relatively large, and this fact will finally lead to a fast descent in the objective. Let us prove (109). It is true when $r = 1$, since by definition we have

$$\mathbf{v}^{(1)} = \mathbf{w}^{(1)} - \mathbf{u}^{(1)} = \upsilon \mathscr{R} \vec{e}, \tag{110}$$

which implies that $\|\psi^{(1)}\| = 0$.

Next, let us assume that (109) is true at the $r$th iteration, we need to prove

$$\psi^{(r+1)} \le 4\mu(r+1)\phi^{(r+1)}, \ \forall \, r < T - 1. \tag{111}$$

To show this result, we utilize (107a) and (107b) to lower and upper bound $4\mu(r+1)\phi^{(r+1)}$ and $\psi^{(r+1)}$, respectively. Substituting (107b) into LHS of (111), we have the upper bound of $\psi^{(r+1)}$, i.e.,

$$\psi^{(r+1)} \le (1 + \epsilon_H \beta) \, 4\mu r \phi^{(r)} + \mu\sqrt{(\phi^{(r)})^2 + (\psi^{(r)})^2}. \tag{112}$$

Applying (107a) into RHS of (111), we have the lower bound of $4\mu(r+1)\phi^{(r+1)}$ as the following:

$$4\mu(r+1)\phi^{(r+1)} \ge 4\mu(r+1)\left( (1 + \epsilon_H\beta)\phi^{(r)} - \mu\sqrt{(\phi^{(r)})^2 + (\psi^{(r)})^2} \right). \tag{113}$$

Next, we will show that the following holds,

$$(1 + 4\mu(r+1)) \left( \sqrt{(\phi^{(r)})^2 + (\psi^{(r)})^2} \right) \le 4\phi^{(r)}. \tag{114}$$

If this is true, then after manipulation, we can show that the RHS of (113) is greater than the RHS of (112), which will eventually imply (111).

In the following, we will show that the above relation (114) is true, i.e., RHS of (107a) is greater than RHS of (107b).

**First step:** We know that

$$4\mu(r+1) \le 4\mu T \stackrel{(108)}{\le} 40\beta L_2 \widehat{c} \mathscr{S} \cdot \mathscr{T} \stackrel{(a)}{\le} \frac{40}{\widehat{c}} \stackrel{(b)}{\le} 1 \tag{115}$$

where the first inequality is true because $r < T - 1$; in $(a)$ we use the relation $\beta L_2 \mathscr{S} \widehat{c} \cdot \mathscr{T} = \frac{1}{\widehat{c}}$ by applying (84b)(84c); $(b)$ is true when $\widehat{c} \ge 40$.

**Second step:** By using the induction assumption and the previous step, we have

$$4\phi^{(r)} \ge 2\sqrt{2(\phi^{(r)})^2} \stackrel{(109),(115)}{\ge} (1 + 4\mu(r+1))\sqrt{(\phi^{(r)})^2 + (\psi^{(r)})^2}, \tag{116}$$

which gives (114). Therefore, we can conclude that $\psi^{(r+1)} \le 4\mu(r+1)\phi^{(r+1)}$ is true, which completes the induction.

**Recursion of $\phi^{(r)}$:** Next we will show that the projection of $\mathbf{v}^r$ on the negative curvature direction $\vec{e}$ will be exponentially increasing.
Using (109), we have

$$\psi^{(r)} \stackrel{(109)}{\le} 4\mu r \phi^{(r)} \stackrel{(115)}{\le} \phi^{(r)}. \tag{117}$$

Then, we can get the recursion of $\phi^{(r+1)}$ by the following steps.

$$\phi^{(r+1)} \stackrel{(107a)}{\ge} (1 + \epsilon_H\beta)\phi^{(r)} - \mu\sqrt{(\phi^{(r)})^2 + (\psi^{(r)})^2} \stackrel{(a)}{\ge} (1 + \epsilon_H\beta)\phi^{(r)} - \mu\sqrt{2}\phi^{(r)}$$

$$\stackrel{(108)}{=} (1 + \epsilon_H\beta)\phi^{(r)} - 10\beta L_2 \mathscr{S}\sqrt{2}\phi^{(r)}$$

$$\stackrel{(84b)}{=} (1 + \epsilon_H\beta)\phi^{(r)} - \frac{10\sqrt{2}\epsilon_H\beta}{\widehat{c}^2 \log(\frac{d\kappa}{\delta})}\phi^{(r)} \stackrel{(b)}{\ge} (1 + \frac{\epsilon_H\beta}{2})\phi^{(r)}$$

where $(a)$ is true because (117); $(b)$ is true when $\widehat{c} \ge 2\sqrt{5\sqrt{2}}$.

**Quantifying Escaping Time:** Next we estimate how many iterations does it require for $\mathbf{w}^{(r)}$ to reduce the objective value sufficiently.

From (105) and the definition of $\phi^{(r)}$, we have

$$6\mathscr{S} \geq \|\mathbf{v}^{(r)}\| \geq \phi^{(r)}$$

$$\overset{(118)}{\geq} (1 + \frac{\beta\epsilon_H}{2})^r \phi^{(1)} \tag{118}$$

$$\overset{(a)}{=} (1 + \frac{\beta\epsilon_H}{2})^r \|\mathbf{w}^{(1)} - \mathbf{u}^{(1)}\| \tag{119}$$

$$\overset{(b)}{\geq} (1 + \frac{\beta\epsilon_H}{2})^r \frac{\delta}{2\sqrt{d}} \frac{\mathscr{S}}{\widehat{c}^2\kappa} \log^{-1}(\frac{d\kappa}{\delta}) \quad \forall r < T \tag{120}$$

where in $(a)$ we used (97); in $(b)$ we use condition $\upsilon \in [\delta/(2\sqrt{d}), 1]$.

Since (120) is true $\forall r < T$, then it must hold for $r = T - 1$. Taking log on both sides of (120), letting $r = T - 1$, we can have

$$T \leq \frac{\log(12\widehat{c}^2(\frac{\kappa\sqrt{d}}{\delta})\log(\frac{d\kappa}{\delta}))}{\log(1 + \frac{\beta\epsilon_H}{2})} + 1 \overset{(a)}{<} \frac{4\log(12\widehat{c}^2(\frac{\sqrt{d}\kappa}{\delta})\log(\frac{d\kappa}{\delta}))}{\beta\epsilon_H} + 1$$

$$\overset{(b)}{<} \frac{4\log(12\widehat{c}^2(\frac{d\kappa}{\delta})^2)}{\beta\epsilon_H} + 1 \overset{(c),(84c)}{\leq} 4(2 + \log(12\widehat{c}^2))\mathscr{T} + 1$$

$$\overset{(d),(84c)}{\leq} 4(2\frac{1}{4} + \log(12\widehat{c}^2))\mathscr{T} \tag{121}$$

where $(a)$ comes from inequality $\log(1 + x) > x/2$ when $x < 1$, in $(b)$ we used relation $\log(x) < x, x > 0$, and $(c)$ is true because $\delta \in (0, \frac{d\kappa}{e}]$ and $\log(d\kappa/\delta) > 1$ so that $\log(12\widehat{c}^2) + 2\log(\frac{d\kappa}{\delta}) \leq (\log(12\widehat{c}^2) + 2)\log(\frac{d\kappa}{\delta})$; $(d)$ is true due to the fact that $\beta L_1 \leq 1$, $\kappa \geq 1$, and $\log(d\kappa/\delta) \geq 1$ so we have $\mathscr{T} \geq 1$.

From (121), we know that when the following holds, we will have $T < \widehat{c}\mathscr{T}$:

$$4\left(2\frac{1}{4} + \log(12\widehat{c}^2)\right) < \widehat{c}, . \tag{122}$$

It can be observed that LHS of (122) is a logarithmic with respect to $\widehat{c}$ and RHS of (122) is a linear function in terms of $\widehat{c}$, implying that when $\widehat{c}$ is large enough inequality (122) holds. It is can be numerically checked that when $\widehat{c} \geq 51$ inequality (122) holds. The proof is complete.

$\square$

### D.3 Convergence Results of SGDN

**Theorem 2.** *Suppose SGDN uses $\beta \leq 1/L_1$, and the following parameters:*

$$T \geq \frac{\widehat{c}\log(\frac{dL_1}{\epsilon_H\delta})}{\beta\epsilon_H} + 1, \quad \mathscr{F} = \frac{\epsilon_H^3}{L_2^2\widehat{c}^5\log^3(\frac{dL_1}{\epsilon_H\delta})}, \quad \mathscr{R} = \frac{\epsilon_H^2}{L_1L_2\widehat{c}^4\log^2(\frac{dL_1}{\epsilon_H\delta})} \tag{123}$$

*where $\widehat{c} \geq 51$. Then, for any $0 < \delta < 1$, $\epsilon_H \leq L_1$, SGDN returns $\diamond$ and a vector $\mathbf{z}$ such that the following holds:*

$$\frac{\mathbf{z}^T\nabla^2 f(\mathbf{x})\mathbf{z}}{\|\mathbf{z}\|^2} \leq -\frac{\epsilon_H}{8\widehat{c}\log(\frac{dL_1}{\epsilon_H\delta})} \tag{124}$$

*with probability $1 - \delta$. Otherwise SGDN returns $\emptyset$ and vector 0, indicating that $\lambda_{\min}(\mathbf{H_P}(\mathbf{x})) \geq -\epsilon_H$ with probability $1 - \delta$.*

The proof of Theorem 2 is similar as the one of proving convergence of PA-GD shown in [21, Lemma 9] or PGD shown in [17, Lemma 14,15] or NEON in [19, Theorem 2] but it is still sufficiently different. Considering the completeness of this paper, here we give the following proof of this theorem in details.

*Proof.* Let $\mathbf{z}^{(1)}$ be a vector that follows uniform distribution within the ball $\mathbb{B}_{\mathbf{x}}^{(d')}(\mathscr{R})$, where $\mathbb{B}_{\mathbf{x}}^{(d')}$ denotes the $d'$-dimensional ball centered at $\mathbf{x}$ with radius $\mathscr{R}$ and $d' = |\mathcal{F}(\mathbf{x})|$.

**Step 1:** We will quantify the decrease of the objective value after $T$ number of iterations. Let $\mathbf{x}$ denote a saddle point which satisfies Condition 1. Consider two sequences generated by SGDN, i.e., $\{\mathbf{u}^{(r)}\}$ and $\{\mathbf{w}^{(r)}\}$, where the initial points of these two sequences satisfy the conditions (97) as shown in Lemma 10.

Again, without loss of generality, we assume $\mathbf{u}^{(1)} = 0$ and let $T^* \triangleq \widehat{c}\mathscr{T}$ and $T' \triangleq \inf_{r \geq 1}\left\{r \mid \widehat{f}(\mathbf{u}^{(r)}) - \widehat{f}(\mathbf{u}^{(1)}) \leq -2\mathscr{F}\right\}$. Then, we have the following two cases to analyze the decrease of the objective value.

**Case $T' \leq T^*$** Applying Lemma 9, we know that

$$f(\mathbf{x} + \mathbf{u}^{(T')}) - f(\mathbf{x}) - \nabla f(\mathbf{x})^T \mathbf{u}^{(T')}$$
$$\leq f(\mathbf{x} + \mathbf{u}^{(1)}) - f(\mathbf{x}) - \nabla f(\mathbf{x})^T \mathbf{u}^{(1)} - 2\mathscr{F} \overset{(a)}{\leq} \frac{L_1}{2}\|\mathbf{u}^{(1)}\|^2 - 2\mathscr{F} \overset{(b)}{\leq} -2\mathscr{F} \quad (125)$$

where $(a)$ is true because of the $L_1$-gradient Lipschitz continuity; $(b)$ is true because $\mathbf{u}^{(1)} = 0$.

From (82) and (57), we know that SGDN always reduces the approximate objective function $\widehat{f}$. When $\widehat{c} \geq 1$ for any $T > \widehat{c}\mathscr{T} = T^* \geq T'$, we have

$$f(\mathbf{x} + \mathbf{u}^{(T)}) - f(\mathbf{x}) - \nabla f(\mathbf{x})^T \mathbf{u}^{(T)} \leq f(\mathbf{x} + \mathbf{u}^{(T^*)}) - f(\mathbf{x}) - \nabla f(\mathbf{x})^T \mathbf{u}^{(T^*)}$$
$$\leq f(\mathbf{x} + \mathbf{u}^{(T')}) - f(\mathbf{x}) - \nabla f(\mathbf{x})^T \mathbf{u}^{(T')} \leq -2\mathscr{F}. \quad (126)$$

Also, since $\mathbf{u}^{(T')} = \mathbf{u}^{(T'-1)} - \beta(\nabla f(\mathbf{x} + \mathbf{u}^{(T'-1)}) - \nabla f(\mathbf{x}))$, we have $\|\mathbf{u}^{(T')}\| \leq \|\mathbf{u}^{(T'-1)}\| + \beta L_1 \|\mathbf{u}^{(T'-1)}\| \leq 4\mathscr{S}$ by $L_1$-gradient Lipschitz continuity, $\beta \leq 1/L_1$ and applying $\|\mathbf{u}^{(r)} - \mathbf{u}^{(1)}\| \leq 2\mathscr{S}$ in Lemma 9.

**Case $T' > T^*$** Applying Lemma 9, we know that $\|\mathbf{u}^{(r)} - \mathbf{u}^{(1)}\| \leq 2\mathscr{S}$ for $r < T^*$. Define $T'' = \inf_{r \geq 1}\left\{r \mid \widehat{f}(\mathbf{w}^{(r)}) - \widehat{f}(\mathbf{w}^{(1)}) \leq -2\mathscr{F}\right\}$. Then, after applying Lemma 10, we know $T'' < T^*$. Using the same argument as the above case, for $T \geq \widehat{c}\mathscr{T} = T^* > T''$, we also have

$$f(\mathbf{x} + \mathbf{w}^{(T)}) - f(\mathbf{x}) - \nabla f(\mathbf{x})^T \mathbf{w}^{(T)} \leq f(\mathbf{w}^{(T^*)}) - f(\mathbf{x}) - \nabla f(\mathbf{x})^T \mathbf{w}^{(T^*)}$$
$$\leq f(\mathbf{w}^{(T'')}) - f(\mathbf{x}) - \nabla f(\mathbf{x})^T \mathbf{w}^{(T'')}) \leq f(\mathbf{w}^{(1)}) - f(\mathbf{x}) - \nabla f(\mathbf{x})^T \mathbf{w}^{(1)} - 2\mathscr{F}$$
$$\overset{(a)}{\leq} \frac{L_1}{2}\|\mathbf{w}^{(1)}\|^2 - 2\mathscr{F} \overset{(b)}{\leq} -1.5\mathscr{F} \quad (127)$$

where $(a)$ is true again due to the $L_1$-gradient Lipschitz continuity; in $(b)$ we use the initialization conditions of the iterates shown in (97) in Lemma 10 so that we have

$$L_1 \mathscr{R}^2 / 2 \overset{(86),(84b)}{=} L_1 \epsilon_H^2 / (2\widehat{c}^8 \kappa^2 L_2^2 \log^4(d\kappa/\delta))$$
$$\overset{(85)}{\leq} \epsilon_H^3 / (2\widehat{c}^8 \kappa L_2^2 \log^3(d\kappa/\delta)) \leq \epsilon_H^3 / (2L_2^2 \widehat{c}^5 \log^3(d\kappa/\delta)) \overset{(84a)}{=} 0.5\mathscr{F}. \quad (128)$$

Also, similar as the previous case, we have $\|\mathbf{w}^{(T'')}\| \leq 4\mathscr{S}$ since $\|\mathbf{w}^{(1)} - \mathbf{x}\| \leq 2\mathscr{R}$.

**Step 2:** We show that at least one sequence, i.e., either $\mathbf{u}^{(r)}$ or $\mathbf{w}^{(r)}$, will give the sufficient descent of the approximate objective value after $T$ iterations. Combining (126) and (127), we have

$$\min\left\{f(\mathbf{x} + \mathbf{u}^{(T)}) - f(\mathbf{x}) - \nabla f(\mathbf{x})^T \mathbf{u}^{(T)}, f(\mathbf{x} + \mathbf{w}^{(T)}) - f(\mathbf{x}) - \nabla f(\mathbf{x})^T \mathbf{w}^{(T)}\right\}$$
$$\leq -1.5\mathscr{F}, \quad \forall T \geq \widehat{c}\mathscr{T}, \quad (129)$$

meaning that at least one of the sequences can give a sufficient decrease of the objective function if the initial points of the two sequences are separated apart with each other far enough along the negative curvature direction $\vec{\mathbf{e}}$.

Let $\mathcal{X}_{\text{stuck}}$ denote the set where a generic sequence $\mathbf{u}^{(r)}$ is initialized such that the sequence cannot escape from the strict saddle point after $T$ iterations, i.e., $f(\mathbf{x} + \mathbf{u}^{(T)}) - f(\mathbf{x}) - \nabla f(\mathbf{x})^T \mathbf{u}^{(T)} >$

$-1.5\mathscr{F}$. According to (129) and Lemma 10, we can conclude that if $\mathbf{u}^{(1)} \in \mathcal{X}_{\text{stuck}}$, then initialization $(\mathbf{u}^{(1)} \pm \upsilon\mathscr{R}\vec{\mathbf{e}}) \notin \mathcal{X}_{\text{stuck}}$ where $\upsilon \in [\frac{\delta}{2\sqrt{d}}, 1]$.

**Step 3:** Next, by leveraging [17, Lemma 14] we give the upper bound of the volume of $\mathcal{X}_{\text{stuck}}$,

$$
\mathsf{Vol}(\mathcal{X}_{\text{stuck}}) = \int_{\mathbb{B}_{\mathbf{x}}^{(d')}} d\mathbf{u} I_{\mathcal{X}_{\text{stuck}}}(\mathbf{u}) = \int_{\mathbb{B}_{\mathbf{x}}^{(d'-1)}} d\mathbf{u}_{-1} \int_{\mathbf{x}_1-\sqrt{\mathscr{R}^2-\|\mathbf{x}_{-1}-\mathbf{u}_{-1}\|^2}}^{\mathbf{x}_1+\sqrt{\mathscr{R}^2-\|\mathbf{x}_{-1}-\mathbf{u}_{-1}\|^2}} d\mathbf{u}_1 I_{\mathcal{X}_{\text{stuck}}}(\mathbf{u})
$$

$$
\leq \int_{\mathbb{B}_{\mathbf{x}}^{(d'-1)}} d\mathbf{u}_{-1} \left(2\frac{\delta}{2\sqrt{d'}\mathscr{R}}\right) = \mathsf{Vol}\left(\mathbb{B}_{\mathbf{x}}^{(d'-1)}(\mathscr{R})\right) \frac{\mathscr{R}\delta}{\sqrt{d'}}
$$

where $I_{\mathcal{X}_{\text{stuck}}}(\mathbf{u})$ is an indicator function showing that $\mathbf{u}$ belongs to set $\mathcal{X}_{\text{stuck}}$, and $\mathbf{u}_1$ represents the component of vector $\mathbf{u}$ along $\vec{\mathbf{e}}$ direction, and $\mathbf{u}_{-1}$ is the remaining $d'-1$ dimensional vector.

Then, the ratio of $\mathsf{Vol}(\mathcal{X}_{\text{stuck}})$ over the volume of the initialization/perturbation ball can be upper bounded by

$$
\frac{\mathsf{Vol}(\mathcal{X}_{\text{stuck}})}{\mathsf{Vol}(\mathbb{B}_{\mathbf{x}}^{(d')}(\mathscr{R}))} \leq \frac{\frac{\mathscr{R}\delta}{\sqrt{d'}}\mathsf{Vol}(\mathbb{B}_{\mathbf{x}}^{(d'-1)}(\mathscr{R}))}{\mathsf{Vol}(\mathbb{B}_{\mathbf{x}}^{(d')}(\mathscr{R}))} = \frac{\delta}{\sqrt{d\pi}}\frac{\Gamma(\frac{d'}{2}+1)}{\Gamma(\frac{d'}{2}+1)} \leq \frac{\delta}{\sqrt{d'\pi}}\sqrt{\frac{d'}{2}+\frac{1}{2}} \leq \delta
$$

where $\Gamma(\cdot)$ denotes the Gamma function, and inequality is true due to the fact that $\Gamma(x+1)/\Gamma(x+1/2) < \sqrt{x+1/2}$ when $x \geq 0$.

**Step 4:** finally, we show that the output of SGDN can give an approximate eigenvector whose smallest eigenvalue is less than $-\epsilon_H$ with high probability. Combining (129) and the results of the last step, we can show that

$$
f(\mathbf{x}+\mathbf{z}^{(T)}) - f(\mathbf{x}) - \nabla f(\mathbf{x})^T\mathbf{z}^{(T)} \leq -1.5\mathscr{F} \tag{130}
$$

with at least probability $1-\delta$. By the $L_2$-Lipschitz continuity, we have

$$
\left| f(\mathbf{x}+\mathbf{z}^{(T)}) - f(\mathbf{x}) - \nabla f(\mathbf{x})^T\mathbf{z}^{(T)} - \frac{1}{2}(\mathbf{u}^{(T)})^T\nabla^2 f(\mathbf{x})\mathbf{u}^{(T)} \right| \leq \frac{L_2}{6}\|\mathbf{z}^{(T)}\|^3. \tag{131}
$$

and $\|\mathbf{z}^{(T)}\| \leq 4\mathscr{S}$. Applying (130) into (131), we have

$$
\frac{1}{2}(\mathbf{u}^{(T)})^T\nabla^2 f(\mathbf{x})\mathbf{u}^{(T)} \leq f(\mathbf{x}+\mathbf{z}^{(T)}) - f(\mathbf{x}) - \nabla f(\mathbf{x})^T\mathbf{z}^{(T)} + \frac{L_2}{6}\|\mathbf{z}^{(T)}\|^3
$$

$$
\overset{(a)}{\leq} -1.5\mathscr{F} + 0.5\mathscr{F} \leq -\mathscr{F} \tag{132}
$$

where in $(a)$ we use (84a)(84b) so that we have $\widehat{c}L_2\mathscr{S}^3 = \mathscr{F}$ where $\widehat{c} \geq 51$. Therefore, we have

$$
\frac{(\mathbf{u}^{(T)})^T\nabla^2 f(\mathbf{x})\mathbf{u}^{(T)}}{\|\mathbf{z}^{(T)}\|^2} \leq \frac{-2\mathscr{F}}{(4\mathscr{S})^2} \overset{(84a),(84b)}{\leq} -\frac{\epsilon_H}{8\widehat{c}\log(d\kappa/\delta)} \tag{133}
$$

so that we can claim that if SGDN returns $\Diamond$ then with probability $1-\delta$ (133) holds for the output $\mathbf{z}^{(T)}$, otherwise SGDN returns $\emptyset$ which indicates that $\lambda_{\min}(\mathbf{H}_{\mathbf{P}}(\mathbf{x})) \geq -\epsilon_H$ with probability $1-\delta$. $\qquad\square$

### D.4 Proof of Corollary 3

*Proof.* The main difference between SNAP and SNAP$^+$ is that we replace the oracle *Negative-Eigen-Pair* by SGDN. Other steps of the convergence analysis are the same as the proof of Theorem 1. Here, we only focus on the difference of the objective reduction between SNAP and SNAP$^+$. First, let the number of iterations run by SGDN for extracting the negative curvature once be

$$
T_{\text{SGDN}} = \frac{\widehat{c}\log(\frac{dL_1}{\epsilon_H\delta})}{\beta\epsilon_H} + 1 \sim \mathcal{O}\left(\frac{\log(1/\epsilon_H)}{\epsilon_H}\right). \tag{134}
$$

In the following, we show the objective reduction in the NCD step, where the number of iterations required in the inner loop is taken into account.

**Case 1) ($\textbf{flag}_\alpha = \emptyset$):** The algorithm implements $\mathbf{x}^{(r+1)} = \mathbf{x}^{(r)} + \alpha^{(r)}\mathbf{d}^{(r)}$ without using $\alpha_{\max}^{(r)}$ computed by (23). By (73), we have the descent of the objective value by $f(\mathbf{x}^{(r+r_{\text{th}})}) \leq f(\mathbf{x}^{(r)}) - \frac{\Delta}{T_{\text{SGDN}}}$.

**Case 2) (flag$_\alpha = \Diamond$):** $\alpha_{\max}^{(r)}$ is computed by (23) to update $\mathbf{x}^{(r+1)}$; By (74), we have $f(\mathbf{x}^{(r)}) - f(\mathbf{x}^{(r-\min\{d,m\})}) < -\min\left\{\epsilon_G^2/(18L_1), 3\epsilon_H'^3(\delta)/(8L_2^2 T_{\mathsf{SGDN}})\right\}, \forall r > \min\{d, m\}$. Applying the same argument from (75) to (80), we know the upper bound of the number of iterations by $\frac{(f(\mathbf{x}^{(1)}) - f^\star)}{\Delta'} \cdot T_{\mathsf{SGDN}}$. From Theorem 2, we know that $\epsilon_H'(\delta) = \frac{\epsilon_H}{8\widehat{c}\log(d\kappa/\delta)}$, i.e., $\gamma = 8\widehat{c}\log(d\kappa/\delta)$. Applying $\epsilon_G = \epsilon$, $\epsilon_H = \sqrt{L_2\epsilon}$ and $\beta \leq 1/L_1$ (note that $r_{\mathsf{th}}$ could be either a constant or chosen in the order of $\mathcal{O}(L_1/\sqrt{L_2\epsilon})$), we can obtain the convergence rate of SNAP$^+$ by (16). $\qquad\square$

# E Numerical Results

In this section, we will provide more numerical results that showcase the strength of SNAP$^+$ and SNAP in applications of solving non-convex machine learning problems.

## E.1 Toy Example

First, we test the algorithms on a toy example where the objective function is constructed by spline functions as the following. Consider function $l(t)$ defined as

$$l(t) = \begin{cases} t^3, & t \in [0, \frac{1}{2}\tau), \\ (t - n\tau)^3 + \frac{1}{4}n\tau^3 & t \in [n\tau - \frac{1}{2}\tau, n\tau + \frac{1}{2}\tau), n = 1, \dots, N, \\ (t - N\tau)^3 + \frac{1}{4}N\tau^3, & t \in [N\tau + \frac{1}{2}\tau, \infty) \end{cases} \tag{135}$$

where $N$ and $\tau$ are some constants. It is clear that the points at $t = L/2$, $t = 1.5L$, ..., are strict saddle points and the one at $t = 0$ is the global optimal solution. Then, we define the objective function as $f(\mathbf{x}) = l((1/d) \sum_{i=1}^{d} \mathbf{x}_i)$, where $\mathbf{x} \in \mathbb{R}^d$. This type of staircase-like function has been widely used in the unconstrained non-convex problems for showing the capability of PGD to escape from saddle points [48, 49]. Here, we choose $\tau = 1$ and $N = 4$, and require $\mathbf{x} \geq 1$ as constraints. Therefore, the optimization problem is $\min_{\mathbf{x} \geq 1} l((1/d)\mathbf{x}^T\mathbf{x})$. Further, we randomly initialize $\mathbf{x}^{(1)} \in \mathbb{R}^{500}$ where each entry follows a uniform distribution in $[0, 4]$, set the step-sizes of PGD, SNAP, and SNAP$^+$ as $\alpha_\pi = 0.1$, take $\pi_{\mathcal{A}^\perp}(\cdot) = \max\{\cdot, 1\}$, and select the hyper-parameters $\epsilon_G = 1 \times 10^{-3}$, $T = 100$, $r_{\text{th}} = 300$ for SNAP and $\beta = 0.01$, $\mathcal{R} = 1 \times 10^{-4}$ and $\mathcal{F} = 100$ for SGDN.

(a) Loss value versus iteration      (b) Loss value versus computational time

Figure 2: The convergence behaviors of SNAP, SNAP$^+$, PGD, and PGD-LS for the staircase-like objective function under linear constraints.

From Figure 2, it can be shown that both SNAP and SNAP$^+$ are able to escape from the saddle points and converge to a lower optimal point compared with both PGD and PGD-LS. Regarding the computational time, SNAP$^+$ converges faster than SNAP since EVD is avoided during the process of exploring the directions of negative curvature.

## E.2 NMF Problems

We also consider the NMF problem, which is

$$\min_{\mathbf{W} \in \mathbb{R}^{n \times k}, \mathbf{H} \in \mathbb{R}^{p \times k}} \|\mathbf{W}\mathbf{H}^T - \mathbf{M}\|^2 \tag{136}$$

$$\text{s.t.} \quad \mathbf{W} \geq 0, \mathbf{H} \geq 0. \tag{137}$$

In particular, we compare the algorithms on both synthetic and real datasets for the MNF problem and show the superiority of exploiting negative curvatures in constrained non-convex problems.

### E.2.1  Synthetic Dataset

The data matrices are randomly generated, where $p = 20$, $n = 50$, $k = 10$, $\mathbf{M} = \mathbf{W}\mathbf{H}^T$, and $[\mathbf{W}; \mathbf{H}] \in \mathbb{R}^{(n+p)\times k}$ follows the uniform distribution in the interval $[0, 1]$. Further, we randomly set 5% entries of $\mathbf{M}$ as 0. The starting point for all the algorithms is $\mathbf{X}^{(1)} = c\pi_{\mathcal{A}^\perp}([\mathbf{W}^{(1)}; \mathbf{H}^{(1)}])$, where $\mathbf{W}^{(1)}$ and $\mathbf{H}^{(1)}$ are randomly generated and follow Gaussian distribution $\mathcal{CN}(0, 1)$ and $\pi_{\mathcal{A}^\perp}(\cdot)$ here is component-wise projection operator $\pi_+(\cdot) = \max\{\cdot, 0\}$. Clearly, the origin point is a strict saddle point. We use three different constants $c$ to initialize sequence $\mathbf{X}^{(r)}$ and the results are shown in Figure 3–Figure 5, where step-size $\alpha_\pi$ chosen for PGD, SNAP, and SNAP$^+$ is 0.01 and the step-size for the gradient based Alt-Min is 0.02, $\beta = 0.01$, $\epsilon_G = 1\times10^{-3}$, $T = 100$, $r_{\text{th}} = 600$, $\mathscr{R} = 1\times10^{-4}$ and $\mathscr{F} = 100$. Note that the stopping criteria are removed in the simulation, otherwise PGD and PGD-LS will not give any output if the initialing point is close to origin.

(a) Loss value versus iteration       (b) Loss value versus computational time

Figure 3: The convergence behaviors of SNAP, SNAP$^+$, PGD, PGD-LS, Alt-Min for NMF, where $c = 1$.

(a) Loss value versus iteration       (b) Loss value versus computational time

Figure 4: The convergence behaviors of SNAP, SNAP$^+$, PGD, PGD-LS, Alt-Min for NMF, where $c = 1 \times 10^{-5}$.

It can be observed that when $c$ is large, all algorithms can converge to the global optimal point of this NMF problem, whereas when $c$ is small as shown in Figure 5 PGD and PGD-LS only converge to a point that has a very large loss value compared with the ones achieved by SNAP and SNAP$^+$. These results show that when the iterates are near the strict saddle points, by exploring the negative curvature, SNAP and SNAP$^+$ are able to escape from the saddle points quickly and converge to the global optimal solutions. Comparing SNAP and SNAP$^+$, we can see that the computational time of SNAP$^+$ is less than SNAP. The reason is simple, which is the computational complexity of calculation of Hessian and eigen-decomposition is too high so that SNAP takes more time to converge. By accessing the gradient and loss value of the objective function, SNAP$^+$ is only required

(a) Loss value versus iteration

(b) Loss value versus computational time

Figure 5: The convergence behaviors of SNAP, SNAP$^+$, PGD, PGD-LS, Alt-Min for NMF, where $c = 1 \times 10^{-10}$.

to compute one eigen-vector whose eigenvalue is the smallest of Hessian around the strict saddle point. The line search algorithm is one of the most effective ways of computing step-sizes. From Figure 3 and Figure 4, it can be observed that PGD-LS converges faster than PGD in terms of iterations but costs more computational time. SNAP and SNAP$^+$ are using line search occasionally rather than each step, so the computational time is not as high as PGD-LS.

### E.2.2 Real Dataset

We also compare the convergence behaviours of the algorithms on USPS handwritten digits dataset [50], where images are $16 \times 16$ grayscale pixels. In Figure 6, we use the $p = 3250$, $n = 256$, $k = 5$. Since the problem size is large, performing eigenvalue decomposition is prohibitive, so we only compare SNAP$^+$, PGD, and PGD-LS, where $\alpha_\pi = 5 \times 10^{-3}$ and $\beta = 5 \times 10^{-3}$.

(a) Loss value versus iteration

(b) Loss value versus computational time

Figure 6: The convergence behaviors of SNAP$^+$, PGD, PGD-LS for NMF, where $c = 1 \times 10^{-10}$.

### E.3 Nonnegative Two Layer Non-linear Neural Networks

In this section, we consider a nonnegative two layer non-linear neural network, which is

$$\min_{\mathbf{W} \in \mathbb{R}^{k \times d}, \mathbf{H} \in \mathbb{R}^{p \times k}} \|\mathbf{W}\sigma(\mathbf{H}^T\mathbf{X}) - \mathbf{Y}\|^2$$

$$\text{s.t.} \quad \mathbf{W} \geq 0, \mathbf{H} \geq 0 \tag{138}$$

where $\sigma(\cdot)$ denotes the activation function. The formulation has a wide applications in regression and learning problems.

In the numerical simulation, the activation function is chosen as sigmoid. Data matrix $\mathbf{X} \in \mathbb{R}^{p \times n}$ is randomly generated which follows uniform distribution in the interval $[0, 1]$, where $n = 100$ denotes the number of samples and $p = 50$ denotes the number of features. Weight matrices $\mathbf{W} \in \mathbb{R}^{k \times d}$ and $\mathbf{H} \in \mathbb{R}^{p \times d}$ are also randomly generated, where $k = 10$ denotes dimension of the output, $d = 15$ is the dimension of the hidden layer. Then, data matrix $\mathbf{Y} \in \mathbb{R}^{k \times n}$ is generated by $\mathbf{Y} = \mathbf{W}\sigma(\mathbf{H}^T\mathbf{X})$. The step-size $\alpha_\pi$ for PGD, SNAP$^+$ is 0.001, $\beta = 0.001$, $r_{\text{th}} = 50$, $T = 50$, $\mathscr{R} = 1 \times 10^{-4}$, $\mathscr{F} = 50$, and $\epsilon_G = 1 \times 10^{-2}$. From Figure 7, it can be observed that SNAP$^+$ can find the stationary points faster than PGD and PGD-LS.

(a) Loss value versus iteration     (b) Loss value versus computational time

Figure 7: The convergence behaviors of SNAP$^+$, PGD, PGD-LS for NNN, where $c = 1$.

### E.4 Symmetric Matrix Factorization over Simplex

In application of topic modelling, the simplex constraint turns out to be essential in modeling (marginal) probability mass functions. In this section, we also consider symmetric matrix factorization over a simplex constraint as the following,

$$\min_{\mathbf{X} \in \mathbb{R}^{n \times k}} \quad \|\mathbf{M} - \mathbf{H}\mathbf{H}^T\|$$
$$\text{s.t.} \quad \mathbf{H} \geq 0, \quad \mathbf{H}^T\mathbf{1} = \mathbf{1}.$$

In the numerical experiments, the data is generated similar as the NMF case, where $n = 100$, $k = 5$ and each column of $\mathbf{H}$ is normalized. We set $\alpha_\pi = 1 \times 10^{-2}$, $T = 100$, $r_{\text{th}} = 100$, $\mathscr{R} = 1 \times 10^{-4}$ and $\mathscr{F} = 100$. From Figure 8, it is interesting to see that three algorithms converge to different objective values. It turns out there would be multiple stationary points around the origin, where SNAP$^+$ finds the lowest one.

### E.5 Penalized NMF

We also consider a penalized version of NMF, i.e.,

$$\min_{\mathbf{W} \in \mathbb{R}^{n \times k}, \mathbf{H} \in \mathbb{R}^{p \times k}} \|\mathbf{W}\mathbf{H}^T - \mathbf{M}\|^2 + \frac{\rho}{2} \sum_i^m \left((\mathbf{1}^T\mathbf{h}_i)^2 - \|\mathbf{h}_i\|^2\right)$$
$$\text{s.t.} \quad \mathbf{W} \geq 0, \mathbf{H} \geq 0$$

where $\mathbf{h}_i$ denotes the columns of $\mathbf{H}$. It has been shown in [51] that this variant of NMF could provide improved clustering accuracy, compared with the classic NMF. Here, we only utilize this formulation to evaluate the performance of the algorithms. In the numerical experiments, we have the similar experimental step-up as the NMF case in E.2.1. The problem size is $p = 100$, $n = 40$ and $k = 5$. We select $\rho = 0.1$, $\alpha_\pi = \beta = 1 \times 10^{-3}$, $T = 100$, $r_{\text{th}} = 20$, $\mathscr{R} = 1 \times 10^{-4}$ and $\mathscr{F} = 100$. It can be observed from Figure 9, SNAP$^+$ converges to the global minimum points of this penalized NMF problem while other ones converge to some points that have relatively large objective values.

Further, we also implement the algorithms on a relatively larger problem, where $p = 2000$, $n = 50$, $k = 5$. In this case, we compare the algorithms by a large initialization, i.e., $c = 1$. It can be observed from Figure 10 that SNAP$^+$ converges faster with respect to the number of iterations.

(a) Loss value versus iteration

(b) Loss value versus computational time

Figure 8: The convergence behaviors of SNAP$^+$, PGD, PGD-LS for matrix factorization under simplex constraints, where $c = 1 \times 10^{-10}$.

(a) Loss value versus iteration

(b) Loss value versus computational time

Figure 9: The convergence behaviors of SNAP$^+$, PGD, PGD-LS for penalized NMF, where $c = 1 \times 10^{-10}$.

(a) Loss value versus iteration

(b) Loss value versus computational time

Figure 10: The convergence behaviors of SNAP$^+$, PGD, PGD-LS for penalized NMF, where $c = 1$.

## Footnotes

[2] For the notations of the free and active space, please see section 4