[Reviews · NeurIPS 2020]

Review 1

Summary and Contributions: In this paper, the authors consider computing approximate Second Order Stationary Points (SOSPs) of problems with the assumption that the objective is generally non-convex, and constrains are linear. The authors then propose two methods to solve this optimization problem. One is to use the Successive Negative-curvature grAdient Projection (SNAP), which is a second-order algorithm. The other one is based on SNAP but only uses first-order information. Both of these two methods are proved to have polynomial per-iteration complexity and global sublinear rates.

Strengths: 1. The technical contributions of this work are solid. Two proposed methods have convergence guarantees. More specifically, the per-iteration complexity of the proposed methods are polynomial-time and they can converge to a good quality with high probability after certain iterations. Especially, the per-iterations are polynomial time is new compared with previous works which have exponential complexity. 2. The problem considered is an generic optimization problem which could be useful in a wide range of real-world applications. 3. The authors conduct experiments on the valuation of these two methods from simulation and real-world datasets. The empirical results show that SNAP and SNAP+ converge faster and better in most cases.

Weaknesses: 1. There is a lack of discussion of how these objectives (especially considered in experiments such as the NMF and Two Layer Non-linear Neural Networks.) can fit into the assumption 1 w.r.t. parameters L1 and L2. 2. The datasets used both in the paper and supplementary are small-scale. There is a lack of performance comparison on the large-scale datasets.

Correctness: Claims and methods are correct. The empirical methodology is also correct.

Clarity: The presentation is clear. The paper is well written.

Relation to Prior Work: Yes. The authors clearly discussed the related work in the Introduction section.

Reproducibility: No

Additional Feedback: To summarize, the authors consider the problem of finding approximate Second Order Stationary Points (SOSPs). Compared with other works, the authors assume that the objective is generally non-convex and constrains are linear. Two methods are designed to solve this optimization problem. Both of these two methods are proved to have polynomial per-iteration complexity and global sublinear rates. In general, the problem is both theoretically and empirically interesting. The presentation is clear and the thectical contributions are solid. However, my only concern is how these proposed methods can efficiently solve the real-world problems in large-scale since the per-iteration complexity is still high. Based on this concern, I have the following questions: Q1. The authors conduct experiments on training nonnegative neural networks in Section E.3. It would be helpful to discuss how the equation (138) satisfies the assumption 1. Any concrete examples of L1 and L2 in this case? Will these parameters be related to the dimensionality and data size? Q2. What is the performance between the proposed method and methods in [1] (Second Order Frank-Wolfe) and [2] (Trust region). Although, there is no constraint considered in these papers, it seems that it can solve the problem at some sense (see Section 4 of [2]). Q3. There are a lot of methods that can solve the NMF problem, it would helpful to make a comparison between these methods and the proposed. Or the authors can make some discussion on this. [1] Nouiehed, Maher, Jason D. Lee, and Meisam Razaviyayn. "Convergence to second-order stationarity for constrained non-convex optimization." arXiv preprint arXiv:1810.02024 (2018). [2] Nouiehed, Maher, and Meisam Razaviyayn. "A trust region method for finding second-order stationarity in linearly constrained non-convex optimization." arXiv preprint arXiv:1904.06784 (2019). After rebuttal: After I read the rebuttal and other reviewers' comments, I think the technical contribution of this paper is solid. I will keep my score unchanged.


Review 2

Summary and Contributions: This paper considers the problem of computing second order stationary points (SOSP) of smooth nonconvex objectives with linear constraint. Two algorithms are proposed, namely, SNAP and SNAP+.

Strengths: I like this paper very much. Finding second order stationary points for general nonconvex problem is NP hard. This paper identify the situations in which finding second order stationary points is easy, and efficient algorithm is proposed and analyzed. Numerical simulations validates the findings. The theoretical findings are solid, and the numerical results are convincing.

Weaknesses: My major concern is the computational costs of checking SOSP1, in which the smallest eigenvalue of the restricted Hessian is required. Here, by "restricted", I mean the space is the null space of A_A(x^*) rather than the whole space. In addition, such an operation is required repeatedly in the algorithm. For large scale problem, the cost of this step can be overwhelming. How to deal with such a problem needs some discussions/investigations.

Correctness: Yes. But I didn't check the proof.

Clarity: Yes. This paper is well organized and the results are clear.

Relation to Prior Work: Yes

Reproducibility: Yes

Additional Feedback:


Review 3

Summary and Contributions: This paper studies escaping from saddle in polytope constrianed nonconvex optimization. When SC condition is satisfied, the rate is polynomial wrt parameters. =========================================== Rebuttal is clear and makes sense. Raised score to 8. BTW two papers about NMF https://arxiv.org/abs/1810.05355 https://arxiv.org/abs/2002.11323

Strengths: This paper proposes the concrete rate and thourough analysis with the constrained escape from saddle problem.

Weaknesses: I don't see major weaknesses, but I have some questions, raised in "additional feedback" section.

Correctness: I have read through the proof and think it's correct (with minor questions that do not hurt). The experiments are reasonable.

Clarity: Yes. Minor questions are below.

Relation to Prior Work: I think they've done great work in literature review which covers the most important papers such as Ge et al, Jin et al, Mokhtari et al., Nouiehed et al. etc. as well as other important related extensions. The logic is very clear and reasonable.

Reproducibility: Yes

Additional Feedback: I'd like to raise my scores if the questions are addressed. 1. Can you compare with this paper as well? Avdiukhin et al., Escaping Saddle Points with Inequality Constraints via Noisy Sticky Projected Gradient Descent, https://opt-ml.org/papers/2019/paper_30.pdf 2. One drawback might be that, when people talk about GD applied to escaping from saddle, they prefer the simplicity of the oracles and similarity with what people have already applied to convex optimization. That's why literature discuss GD with perturbation or trust region, so the Hessian estimation or line search steps here might weaken that (note the above paper's algo is fairly simple as well). I believe that due to the hardness of constraint some sophistication can be necessary, but I'd suggest trying to make the algorithm simple, or discuss about its simplicity. 3. I guess the SC condition is required everywhere, if so the author should mention it in all the theorems. 4. Due to the SC condition, the result can be limited and is motivated by data driven problem (so randomness guarantees the condition). But I'm wondering if such second order stationaries are really the solution people search for (i.e., what are the qualities of the optimizers)? People have shown that many problems such as matrix factorization are strict saddle functions and have no spurious local minimum, thus unconstrained escaping from saddle works. Is there any instance that the quality of second order minimum is guaranteed in constrained regime (say all local mins are global)? But I think theoretically GD cannot do anything with local min and it's still of value just to point out the convergence rate. 5. Some algorithms are in the appendix, so if they're quoted in the main paper, make sure to say "see appendix section ..." 6. State/Refer to how to choose v in Algorithm 1 line 7. 7. Question: to discuss the active constraint, the iterate has to be exactly on the boundary/hypersurface of the constraint polytope. Is it caused by exactly solving (23) in algorithm 2? Seems proximal operation or projection are not used in the algorithms. 8. A minor comment (doesn't hurt to ignore this): to be more inclusive, the authors can discuss the case when some active constraints are linearly dependent, or MFCQ. 9. (Minor comment) Prop 2,3: I'm not sure why the authors have to discuss \epsilon = 0 or limit case. Is it required for the analysis or practical instances? 10. Lemma 7: "Algorithm 1 will stop..." is it referred to algorithm 1 or only NCD step? And in algorithm 1, once a constraint turns active, is it always active in the following iterates?

[Author Response · NeurIPS 2020]

We thank the reviewers for their thoughtful comments and positive feedback.

**Reviewer 1**. **Q1**: Thank you very much for raising this valid technical point. To satisfy
assumption 1, we can introduce box constraints on the entries of $\mathbf{H}$ and $\mathbf{W}$. Our algorithm
will still be applicable in the presence of the box constraints. In addition, our experiments
show that if the box constraints are chosen large enough, they will not become active over
the iterates of the algorithms and hence will not change the trajectory of the algorithm. We
will add this discussion to the paper and share our code in the presence of constraints.

Figure 1: See Sec. E2.1 for the details of the this experiment ($c = 10^{-5}$).

**Q2**: As mentioned in [1] and [2], solving the subproblems is NP-hard in the number of
constraints for linear constrained problems. Thus, even for a simple NMF problem, solving
subproblems in [1, 2] requires exponential computational complexities in the number of
constraints (which is the same as the number of variables in NMF).

**Q3**: There are different algorithms leveraging the NMF problem structure such as alternating minimization (Alter-
Min). Here, we further compare Alter-Min with SNAP (shown in Figure 1). This plot shows that Alter-Min behaves
similar to PGD-LS (but it costs less computational time numerically than PGD-LS due to its simplicity of the subprob-
lems). We plan to add more numerical experiments and include large-scale cases in the revised version.

**Reviewer 2**. Regarding the computational cost of checking SOSP1, first notice that the number of times we check
SOSP1 condition is reduced in our algorithm by introducing $\mathsf{flag}_\alpha$. When $\mathsf{flag}_\alpha = \emptyset$, we do not check SOSP1
condition for at least $r_{\text{th}}$ iterations. Second, to reduce the computational cost of checking SOSP1, we proposed SGDN
(a first-order method) for SNAP$^+$ with only $\mathcal{O}(d)$ per-iteration computational complexity (instead of $\mathcal{O}(d^2)$ required
for other methods such as Lanczos). SGDN approximates the *smallest* eigenvalue in the interested subspace and avoids
the evaluation of the restricted Hessian matrix. Our numerical experiments demonstrate that the gain obtained by
SGDN procedure outweighs its computational cost. Is particular, in most cases, SNAP$^+$ outperforms classic projected
gradient descent in terms of computational time.

**Reviewer 3**. **Q1**: Yes, we will add the comparison. **Q2**: The line search is a simple step-size selection method with
only logarithmic time complexity. Also notice that our algorithm SNAP$^+$ does not require Hessian estimation and it
only works with gradient. We will include more high level discussions to simplify the understanding of our algorithm.
Thank you for pointing this out.

**Q3**: The SC condition is only used to establish the connection between SOSP1 and SOSP2. In other words, if the
SC condition holds, then the obtained SOSP1 by our algorithms is also a SOSP2. However, our algorithms compute
SOSP1 without needing SC condition to hold.

**Q4**: Showing the global optimality of SOSPs (in some practical problems) is a very interesting future research direc-
tion. However, even when they are not globally optimal, there is still value in computing SOSPs. This is because
SOSPs satisfy stronger optimality conditions and are in general a subset of FOSPs. Hence they have higher chance
of being globally optimal. Moreover, one can always run his/her own favorite algorithm for finding FOSPs and use
our algorithm for escaping saddle points if needed. In addition, our *SOSP-finding* algorithms are "hessian-aware"
and hence they can escape saddle points much faster as demonstrated in our numerical experiments. These benefits
of algorithms developed for finding SOSPs are the main motivation behind many research works in the optimization
society for finding SOSPs. Classical algorithms such as Newton, trust-region, or cubic regularization methods, were
all (at least partially) motivated by these facts (long before researchers showing the global optimality of SOSPs for
certain problem instances).

**Q5**: Thanks for pointing out this issue. We will revise accordingly. **Q6**: If $q_\pi(\mathbf{x}^{(r)})^T\mathbf{v}(\mathbf{x}^{(r)}) > 0$, then we only need to
add a minus sign to $\mathbf{v}(\mathbf{x}^{(r)})$, otherwise, just keep it. We will revise to clarify this point further. **Q7**: Yes. The projection
step is used in line 18 algorithm 1. **Q8**: Good suggestion! We will include a discussion on the LICQ and/or MFCQ in
the revised version.

**Q9**: [Prop. 2] The definitions of the exact SOSP1 and SOSP2 have been used in the literature [35], but the relation
between these two notations is unknown. Here, we provide rigorous proof to show their equivalence under the SC
condition. [Prop. 3] This proposition shows that the introduced notion is continuous and hence proposed algorithms
for computing this stationary notion can indeed escape (strict) saddle points asymptotically. Notice that as explained
in [31, Remark 2.4] the continuity of the measure is necessary for escaping saddle points (where some previous works
could get stuck in a strict saddle point)

**Q10**: i) It refers to Algorithm 1. We will clarify it in our revision. In the case the inactive set becomes empty and
the first-order stationary condition has been satisfied, which implies that Algorithm 1 has already achieved the SOSP1
and therefore will stop. ii) No. Even when a constraint turns active, algorithm 1 is still possible to use PGD to update
the iterates (when the conditions shown in line 3 of algorithm 1 are not satisfied), then the active constraint might be
deactivated after the PGD step.

[Meta-Review · NeurIPS 2020]

All the reviewers were positive towards the paper. Originally, the paper got 687, with relatively high confidences. The major concern about the paper was that it may not fit for large scale problems. During discussion, Reviewer #3 deemed that the rebuttal is "clear and makes sense", hence raised his/her score from 7 to 8. The AC deemed that the theoretical contribution of the paper is good and agreed to forgive the weakness in experiments. Thus the AC recommended acceptance.